# FlowMoE: A Scalable Pipeline Scheduling Framework for Distributed Mixture-of-Experts Training

**Yunqi Gao**[1], **Bing Hu**[1*], **Mahdi Boloursaz Mashhadi**[2], **A-Long Jin**[3], **Yanfeng Zhang**[4],
**Pei Xiao**[2], **Rahim Tafazolli**[2], **Mérouane Debbah**[5]

[1]School of Information Science and Electronic Engineering, Zhejiang University
[2]5GIC & 6GIC, Institute for Communication Systems (ICS), University of Surrey
[3]School of Advanced Technology, Xi'an Jiaotong-Liverpool University
[4]School of Computer Science and Engineering, Northeastern University
[5]KU 6G Research Center, Department of Computer and Information Engineering, Khalifa University
{gaoyunqi1999, binghu}@zju.edu.cn,
{m.boloursazmashhadi, p.xiao, r.tafazolli}@surrey.ac.uk,
along.jin@xjtlu.edu.cn, zhangyf@mail.neu.edu.cn, merouane.debbah@ku.ac.ae

## Abstract

The parameter size of modern large language models (LLMs) can be scaled up via the sparsely-activated Mixture-of-Experts (MoE) technique to avoid excessive increase of the computational costs. To further improve training efficiency, pipelining computation and communication has become a promising solution for distributed MoE training. However, existing work primarily focuses on scheduling tasks within the MoE layer, such as expert computing and all-to-all (A2A) communication, while neglecting other key operations including multi-head attention (MHA) computing, gating, and all-reduce communication. In this paper, we propose FlowMoE, a scalable framework for scheduling multi-type task pipelines. First, FlowMoE constructs a unified pipeline to consistently scheduling MHA computing, gating, expert computing, and A2A communication. Second, FlowMoE introduces a tensor chunk-based priority scheduling mechanism to overlap the all-reduce communication with all computing tasks. We implement FlowMoE as an adaptive and generic framework atop PyTorch. Extensive experiments with 675 typical MoE layers and four real-world MoE models across two GPU clusters demonstrate that our proposed FlowMoE framework outperforms state-of-the-art MoE training frameworks, reducing training time by 13%-57%, energy consumption by 10%-39%, and memory usage by 7%-32%. FlowMoE's code is available at https://github.com/ZJU-CNLAB/FlowMoE.

## 1 Introduction

Large language models (LLMs) have demonstrated remarkable performance on natural language processing (NLP) tasks as model sizes increase (e.g., GPT-3 [1] with 175 billion parameters, LLaMA3.1 [2] with 405 billion parameters, and DeepSeek-R1 [3] with 671 billion parameters). However, parameter scaling causes a linear increase in the computational cost. Currently, sparsely-activated LLMs with Mixture-of-Experts layers (MoE models) have become extremely popular [4], where the MoE layer replaces the standard feed-forward layer in traditional transformer blocks. In MoE, a gating function selects a small subset of dense layers, known as experts, to be activated for the input sample tokens. This dynamic selection enables only a few experts to participate in computation during each iteration [5, 6]. Therefore, MoE techniques can scale the model size with limited increase in computation [5, 7, 8], e.g., Google's Switch Transformer expands parameters from a few billion to

---

*Corresponding author

Table 1: Time for different tasks of each iteration in training four MoE models on a 16-GPU (NVIDIA RTX3090) cluster (with 100Gbps bandwidth) running vanilla expert parallelism [19]. 'MHA + Gating Time' indicates the computing time of the MHA layer and the gating function. 'All-Reduce Time' indicates the time of all-reduce communication. 'Ratio' indicates the ratio of the sum of 'MHA + Gating Time' and 'All-Reduce Time' over the total time per iteration.

| Model | MHA + Gating Time | All-Reduce Time | Iteration Time | Ratio |
|---|---|---|---|---|
| GPT2-Tiny-MoE | 23.5ms | 32.6ms | 169.5ms | 33.1% |
| BERT-Large-MoE | 61.9ms | 98.3ms | 537.8ms | 29.8% |
| LLaMA2-MoE | 308.4ms | 368.8ms | 1987.7ms | 34.2% |
| DeepSeek-V2-S | 870.2ms | 1247.8ms | 5843.3ms | 36.1% |

1.5 trillion through 15 MoE layers with 2048 experts each [9]. However, training MoE models on large-scale GPU clusters still suffers from serious scalability bottlenecks [10, 11, 12].

Recently, expert parallelism has been proposed to train MoE models in a distributed fashion by placing different experts on multiple workers since a single worker (e.g., GPU) cannot hold a complete MoE model [5]. In each iteration, input tokens need to be transferred to particular workers, which depends on an all-to-all (A2A) communication operation (*dispatch*), and the outputs from expert computing on different workers need to be collected via another A2A operation (*combine*) [13]. In addition, the parameters of the remaining parts of the model including the multi-head attention (MHA) layer and the gating function are replicated across all workers to perform data parallelism mode [14, 15, 16, 17, 18]. In particular, the parameters of the MHA layer and the gating function need to be synchronized among all workers using the all-reduce communication after backward propagation.

Since the computing and communication tasks do not occupy the same hardware resources, pipelining of computing and communication becomes one of the most efficient methods for accelerating expert parallelism. Existing works (e.g., ScheMoE [10], Tutel [12], FasterMoE [11], Lina [20], PipeMoE [21], Comet [22]) pipeline expert computing tasks and A2A communication tasks to hide the communication and reduce the training time of the MoE model by splitting the input data of the MoE layer into micro-batches. However, they focus only on overlapping tasks in the MoE layer and neglect MHA layer computing, gating, and all-reduce communication. We conducted experiments on a 16-GPU cluster and the results are shown in Table 1 (see Table 2 for model configurations). It can be observed that the MHA layer computing, gating, and all-reduce communication constitute 30%-40% of the iteration time. Therefore, designing a pipeline scheduling method that considers all major tasks of the transformer block with MoE layer can maximize the overlap between computing and communication, thus improving the scaling efficiency of distributed MoE training.

In this paper, we propose FlowMoE, a scalable pipeline scheduling framework for multi-type tasks in distributed MoE training. We make the following main technical contributions: (1) We construct a unified pipeline that consistently schedules MHA computing, gating, expert computing, and A2A communication. (2) We design a priority-based scheduling mechanism for communication tasks using all-reduce tensor chunks to further overlap the all-reduce communication with computing tasks, and we leverage Bayesian optimization (BO) to automatically tune the partition size of all-reduce chunks. (3) We implement the FlowMoE framework on PyTorch [23] and open-source the code. Extensive experiments on two GPU clusters using manually customized MoE layers and real-world transformer-based MoE models show that FlowMoE achieves better training performance than state-of-the-art MoE frameworks (including ScheMoE [10], FSMoE [24], Tutel [12] and FasterMoE [11]) and vanilla expert parallelism [19]. Specifically, FlowMoE outperforms ScheMoE by 26% average time efficiency in training 675 MoE layers with different configurations. In comparison with state-of-the-art frameworks in training real-world popular MoE models, FlowMoE achieves $1.13\times$-$1.82\times$ speedup, and reduces energy consumption by 10%-41% and memory usage by 7%-32% during each training iteration. The main differences between FlowMoE and the key literature is shown in Appendix A.

## 2 Background and Challenges

### 2.1 Transformer Block with MoE Layer

Fig. 1a illustrates a typical transformer structure with MoE layers, where the transformer block usually consists of an MHA layer and an MoE layer. For the input tensor $I \in \mathbb{R}^{B \times N \times M}$, the MHA

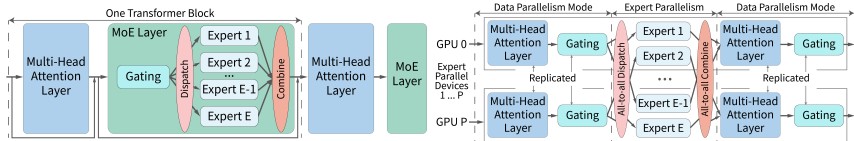

(a) A transformer structure with MoE layers.  (b) An illustration of expert parallelism.

Figure 1: An example of a transformer block with MoE layer and expert parallelism.

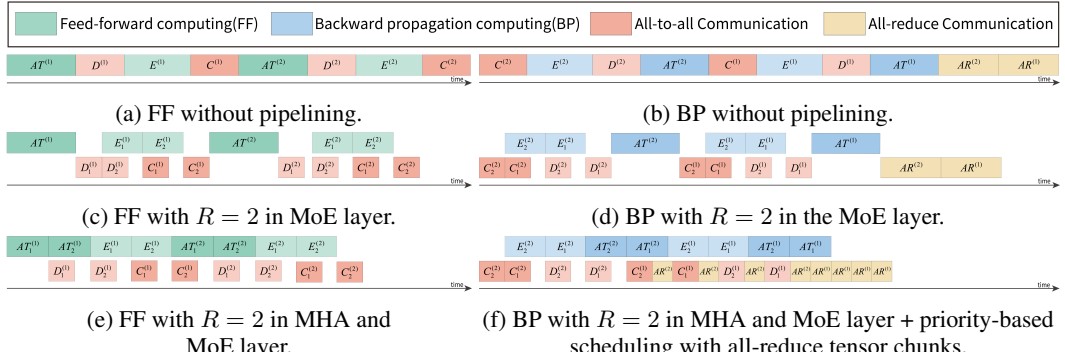

(a) FF without pipelining.

(b) BP without pipelining.

(c) FF with $R = 2$ in MoE layer.

(d) BP with $R = 2$ in the MoE layer.

(e) FF with $R = 2$ in MHA and MoE layer.

(f) BP with $R = 2$ in MHA and MoE layer + priority-based scheduling with all-reduce tensor chunks.

Figure 2: An example of the execution timelines of the computing and communication tasks for a model consisting of two transformer blocks with MoE layer.

layer utilizes the Query, Key, and Value matrices $W^Q, W^K, W^V \in \mathbb{R}^{M \times M}$ to calculate the attention data of each token and obtains the output tensor $I' \in \mathbb{R}^{B \times N \times M}$ through a linear transformation matrix $W^O \in \mathbb{R}^{M \times M}$, where $B$ denotes the number of samples per GPU (or mini-batch size) in one iteration, $N$ denotes the number of tokens per sample, and $M$ denotes the embedding size of a token. The MoE layer includes a *gating function* and multiple *experts*. The gating function $G$ is a small learnable neural network followed by a softmax layer and is used to select the activated experts for the input tokens. Typically, only top-$k$ experts are chosen to process a token [9]. The output tensor $G(I') \in \mathbb{R}^{E \times C \times M}$ of the gating function will be dispatched to the corresponding expert (each expert receives a tensor of shape $C \times M$), where $E$ denotes the total number of experts per MoE layer and $C$ denotes the maximum number of tokens assigned to one expert. $C$ can be calculated by $f \times k \times B \times N/E$, where $f$ is the capacity factor that determines the maximum number of tokens assigned to each expert and is used to adjust $C$ [5]. An expert is usually a small neural network with two structurally symmetric feed-forward layers (the first and the second layers are of sizes $M \times H$ and $H \times M$, respectively), where $H$ denotes the hidden size of the feed-forward layer, and each expert is considered to have its own domain of expertise. After expert computing, the outputs from all the experts are combined into a tensor with a shape of $B \times N \times M$ as the input to the next transformer block.

## 2.2 Expert Parallelism

We define $P$ as the number of workers (or GPUs) in the cluster, $L$ as the number of transformer blocks in a MoE model. Fig. 1b shows an example of training an MoE model on $P$ workers by expert parallelism, where each worker has different expert parameters. Then, $AT_r^{(l)}$, $E_r^{(l)}$, $D_r^{(l)}$, and $C_r^{(l)}$ respectively denote the $r$th subtask of MHA layer (including gating function), expert computing, dispatch A2A communication, and combining A2A communication of the $l$th transformer block. $AR^{(l)}$ denotes all-reduce communication task of the $l$th transformer block. Figs. 2a and 2b show the timelines of multiple computing and communication tasks in the feed-forward computing and backward propagation for one transformer block in vanilla expert parallelism [19]. It is worth noting that the timeline for backward propagation is opposite to that of feed-forward computing, but the parameters of the MHA layer and the gating function need to be synchronized through additional all-reduce communication operation.

## 2.3 Performance Bottlenecks in Distributed MoE Training

Most of existing pipeline scheduling works (e.g., ScheMoE [10], Tutel [12], PipeMoE [21]) partitions the input token tensor of the MoE layer in the data dimension according to the pipelining degree $R$, where Figs. 2c and 2d illustrate a pipelining example with $R = 2$ in the MoE layer of a

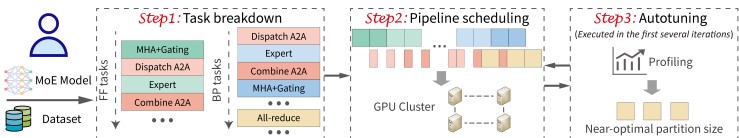

Figure 3: Workflow of FlowMoE.

transformer block. In addition, FasterMoE [11] splits the input tensor of the MoE layer based on the number of workers, enabling point-to-point communication between workers. Although these works effectively reduce the training time by pipelining the expert computing task and the A2A communication task, they neglect the MHA layer computing task, gating task and all-reduce communication task. Moreover, some MoE frameworks (e.g., FSMoE [24] and Lina [20]) also explore the pipeline of all-reduce communication tasks. Nevertheless, FSMoE focuses more on inter- and intra-node communication overlaps within the MoE layer, while Lina only optimizes MoE-layer-specific communication bottlenecks and not the entire transformer block. Therefore, we aim to pipeline all major computing and communication tasks across the transformer block to minimize the per-iteration time, which involves three main challenges:

**Complex dependencies between multi-type tasks.** In distributed MoE training, there exists highly complex dependencies between the computing and communication tasks (see Figs. 2a and 2b) [10]. In particular, the typical parallelization schemes (e.g., pipeline parallelism [25, 26], tensor parallelism [27, 28, 29]) are difficult to be overlaid directly on expert parallelism.

**Coexistence of A2A communication and all-reduce communication.** Although there has been extensive studies [30, 16, 31, 15, 32, 33, 34] providing efficient scheduling algorithms for data-parallel training of all-reduce communication tasks, they focus on traditional convolution-based deep neural networks or LLMs without MoE layers. These studies cannot be directly applied to distributed MoE training because the additional A2A communication tasks are not equivalent to all-reduce communication tasks (see Fig. 2b).

**Designing of an adaptive and generic pipeline scheduling framework.** Currently, the performance of most scheduling frameworks depends on some hyperparameters setting [35, 36, 37], which introduces additional workload for tuning them before training. Ideally, an adaptive framework should automatically tune hyperparameters and be directly deployable by modifying only the model definition or dataset interface. Meanwhile, a general framework should also be designed to be compatible with different optimization frameworks and communication stacks.

In this paper, our main goal is to address the above three challenges by proposing a scalable pipeline scheduling framework called FlowMoE for multi-type tasks in distributed MoE training. All frequently used notations are summarized in Table A.1 of the Appendix.

## 3 FlowMoE

### 3.1 Overview

Fig. 3 illustrates the workflow of FlowMoE. FlowMoE performs efficient pipeline training leveraging the given MoE structure and the dataset. First, multi-type computing and communication tasks are broken down according to the pipelining degree and wait for pipeline scheduling. Second, FlowMoE schedules all computing and communication tasks according to their dependencies and defined priority for distributed training on the GPU cluster (see details in Secs. 3.2 and 3.3). Third, FlowMoE automatically tunes the partition size of all-reduce tensor based on the first several iterations during training by BO profiling (see details in Sec. 4.1).

### 3.2 Pipeline for MHA and MoE Layer

Based on the pipeline of the MoE layer, FlowMoE first pipelines the MHA layer computing and gating. It is worth noting that for the $l$th transformer block, the gating task only depends on the computing task of the MHA layer, and in the subsequent expression, we consider the computing task of both as a whole and denote it by $AT^{(l)}$ ($1 \leq l \leq L$). FlowMoE partitions the input tensor of each transformer block into $R$ equal-sized parts, and each computing or communicating task (except all-reduce communicating task) in the MHA layer and the MoE layer will be partitioned into $R$ independent subtasks. In particular, the tasks with the same type have the same execution time. The partitioned tasks can be represented as the set

$$\mathbb{T} = \left\{ AT_r^{(l)}, D_r^{(l)}, E_r^{(l)}, C_r^{(l)}, AR^{(l)} | 1 \leq r \leq R \right\}, \tag{1}$$

where $AT_r^{(l)}$ and $E_r^{(l)}$ are computing tasks, and $D_r^{(l)}$, $C_r^{(l)}$ and $AR^{(l)}$ are communication tasks.

Then, for $1 \leq l < L$, during feed-forward computing, the scheduling order of computing tasks can be denoted as

$$AT_1^{(l)}->AT_2^{(l)}->...->AT_R^{(l)}->E_1^{(l)}->E_2^{(l)}->...->E_R^{(l)}->AT_1^{(l+1)}->...->E_R^{(l+1)}, \quad (2)$$

the scheduling order of A2A communication tasks can be denoted as

$$D_1^{(l)}->D_2^{(l)}->...->D_R^{(l)}->C_1^{(l)}->C_2^{(l)}->...->C_R^{(l)}->D_1^{(l+1)}->...->C_R^{(l+1)}. \quad (3)$$

During backward propagation, the scheduling order of computing tasks can be denoted as

$$E_R^{(l+1)}->...->AT_1^{(l+1)}->E_R^{(l)}->E_{R-1}^{(l)}->...->E_1^{(l)}->AT_R^{(l)}->AT_{R-1}^{(l)}->...->AT_1^{(l)}, \quad (4)$$

the scheduling order of A2A communication tasks can be denoted as

$$C_R^{(l+1)}->...->D_1^{(l+1)}->C_R^{(l)}->C_{R-1}^{(l)}->...->C_1^{(l)}->D_R^{(l)}->D_{R-1}^{(l)}->...->D_1^{(l)}. \quad (5)$$

Figs. 2e and 2f show the example of execution timelines of the computing and communication tasks when pipelining the MHA layer and the MoE layer with $R = 2$. Compared to pipelining the MoE layer only, the MHA layer computing task and gating task can be overlapped with the A2A communication tasks, thus shortening the per-iteration time.

### 3.3 Pipeline for All-reduce Communication

Existing state-of-the-art scheduling frameworks centrally executes all-reduce communication tasks at the end of backward propagation for each iteration (we call it centralized scheduling of all-reduce communication tasks). To reduce all-reduce communication time, we further consider the pipeline of all-reduce communication tasks. Intuitively, in the backward propagation, the all-reduce communication task of transformer block $l$ can overlap with computing tasks of transformer block $l-1$, since they both depend on the completion of MHA layer computing tasks of transformer block $l$. However, in actual training, the all-reduce communication task of transformer block $l$ will conflict with the A2A communication tasks of transformer block $l-1$. Therefore, we first mathematically model the timeline of backward propagation in one iteration to find the optimal scheduling of the two communication tasks that minimizes the training time. Considering resource competition in real training environments, we assume that only computing and communication tasks can be executed simultaneously on a GPU, while multiple computing or multiple communication tasks cannot be run simultaneously. Furthermore, there is no preemption between tasks: once a task starts execution, it must run to completion without interruption. We define $\tau_b(\cdot)$ as the beginning execution timestamp of a task during the backward propagation, and $t_b(\cdot)$ as the elapsed time of a task during the backward propagation. According to Fig. 2b, the objective function and the dependencies between tasks can be expressed as follows ($1 \leq r \leq R$):

$$\min \quad T_b = \tau_b(AR^{(1)}) + t_b(AR^{(1)}) - \tau_b(C_R^{(L)}) \quad (6)$$

$$\text{s.t.} \quad \tau_b(C_r^{(l-1)}) \geq \tau_b(AT_r^{(l)}) + t_b(AT_r^{(l)}), 1 < l \leq L, \quad (6a)$$

$$\tau_b(E_r^{(l)}) \geq \tau_b(C_r^{(l)}) + t_b(C_r^{(l)}), 1 \leq l \leq L, \quad (6b)$$

$$\tau_b(D_r^{(l)}) \geq \tau_b(E_r^{(l)}) + t_b(E_r^{(l)}), 1 \leq l \leq L, \quad (6c)$$

$$\tau_b(AT_r^{(l)}) \geq \tau_b(D_r^{(l)}) + t_b(D_r^{(l)}), 1 \leq l \leq L, \quad (6d)$$

$$\tau_b(AR^{(l)}) \geq \tau_b(AT_r^{(l)}) + t_b(AT_r^{(l)}), 1 \leq l \leq L. \quad (6e)$$

We use the sign $*$ to denote the timeline for centralized scheduling of all-reduce communication tasks. Now, let $T_b$ denote the backward propagation time when the all-reduce communication task of a transformer block is inserted between any two A2A communication tasks. Also denote by $T_b^*$ the backward propagation time under centralized scheduling of all-reduce communication tasks. With this notation, we have the following theorem:

**Theorem 1.** *If the scheduling order satisfies Eqs. 4 and 5, then we have $T_b \leq T_b^*$.*
*Proof.* Theorem 1 proof is provided in Appendix B. $\qquad\square$

According to Theorem 1, we find that partitioning and inserting all-reduce communication tasks into the gaps between any of the A2A communication tasks can increase the overlap between all-reduce communication tasks and computing tasks. Thus, we design a priority scheduling mechanism of communication tasks based on the all-reduce tensor chunks in FlowMoE. Specifically, during the backward propagation, FlowMoE first slices the tensor for the all-reduce communication tasks of

**Algorithm 1** Training process in FlowMoE

**Input:** Dataset $D = \{(x_1, y_1), ..., (x_n, y_n)\}$, $L$.
**Output:** Model Weights $W = \{W_1, ..., W_L\}$.
1: Initialize DataQueue for data transferred between tasks;
2: Initialize A2AQueue for A2A communication tasks;
3: Initialize ARQueue for all-reduce communication tasks;
4: Split $D$ for $d_1, d_2, ..., d_R$ into DataQueue;
5: Communication_pool_management.start();
6: **for** $l = 1 \rightarrow L$ **do**//Feed-forward Computing
7:    **for** $r = 1 \rightarrow R$ **do**
8:       $d_r$=AT.FFcomp(DataQueue.get());
9:       A2AQueue.put($d_r$);
10:    **for** $r = 1 \rightarrow R$ **do**
11:       $d_r$=E.FFcomp(DataQueue.get());
12:       A2AQueue.put($d_r$);
13: **for** $l = L \rightarrow 1$ **do**//Backward Propagation
14:    **for** $r = 1 \rightarrow R$ **do**
15:       A2AQueue.put(DataQueue.get());
16:    **for** $r = 1 \rightarrow R$ **do**
17:       $d_r$=E.BPcomp(DataQueue.get());
18:       A2AQueue.put($d_r$);
19:    **for** $r = 1 \rightarrow R$ **do**
20:       $d_r$=AT.BPcomp(DataQueue.get());
21:       DataQueue.put($d_r$);
22: Waiting until all-reduce communication is finished;
23: Update $W$;

**Algorithm 2** Communication pool management

1: /* Partition an all-reduce tensor and enqueue tensor chunks. */
2: Get $S_p$ from BO;
3: **procedure** PARTITION($ARTensor$)
4:    $ARChunks = ARTensor$.partition($S_p$);
5:    ARQueue.put($ARChunks$);
6: /* Communication task priority scheduling. */
7: **procedure** COMMPOOLMANAGER
8:    **while** True **do**
9:       **if** A2AQueue is not empty **then**
10:          $d$= A2A.Comm(A2AQueue.get());
11:          DataQueue.put($d$);
12:       **else if** ARQueue is not empty **then**
13:          AR.Comm(ARQueue.get());

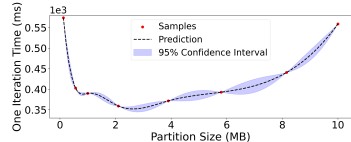

Figure 4: BO example: tuning $S_p$ for training BERT-Large-MoE with a 16-GPU cluster.

each layer into tensor chunks with size $S_p$. Second, FlowMoE maintains a communication task pool which includes all the all-reduce tensor chunks and A2A communication tasks, and sets the scheduling priority of all-reduce tensor chunks to be lower than that of A2A communication tasks. In other words, the communication tasks of the all-reduce tensor chunks will be executed immediately when there is no A2A communication tasks. Fig. 2f shows an example of execution timeline when using 2-degree pipelining in MHA layer and MoE layer as well as priority scheduling mechanism of communication tasks based on all-reduce tensor chunks. It can be observed that the all-reduce tensor chunks fully occupy the gap between the A2A communication tasks and thereby maximize the overlap between the computing tasks and the communication tasks.

## 4 Parameter Optimization, Algorithm Design and System Implementation

### 4.1 Auto-Tuning Partition Size by Bayesian Optimization

Ideally, if the communication of all-reduce tensor chunks does not introduce extra startup overhead, we have the following theorem:

**Theorem 2.** *When the scheduling order satisfies Eqs. 4 and 5, and using the priority scheduling mechanism of communication tasks based on all-reduce tensor chunks, the time per iteration will be minimized if $S_p \rightarrow 0$ and there is no startup overhead for communicating all-reduce tensor chunks.*

*Proof.* Theorem 2 proof is provided in Appendix C. □

In the actual training, the communication tasks of all-reduce tensor chunks will introduce extra startup overhead [15]. Therefore, the partition size of all-reduce tensor chunks, $S_p$, becomes the knob that determines the trade-off between optimal scheduling and system overhead. Since it is non-trivial to explicitly model the iteration time as a function of $S_p$, to guarantee the adaptivity of FlowMoE, we adopt BO to automatically tune $S_p$ during training, which attempts to find good parameters of an unknown objective function in as few number of trials as possible. BO's parameter settings, importance, and performance evaluation are described in detail in Appendix D. The goal of BO in FlowMoE is to minimize the per-iteration time. Specifically, BO fits an objective function by sampling different ($S_p$, per-iteration time) pairs and continuously suggests the next $S_p$ to obtain a new objective function value, where the per-iteration time corresponding to different $S_p$ in all pairs is measured from the average of multiple iterations (e.g., 10 iterations). In other words, BO can accurately predict a near-optimal $S_p$ based on enough samples. Fig. 4 illustrates an example of using BO to tune $S_p$. In the actual training, with only 8 samples, BO returns a near-optimal value at 2.5MB with a good confidence.

Table 2: Benchmark models.

| MoE Model | # Params (MHA+Gating) | # Params (Experts) | Dataset | L | B | N | M | H | E/P | k | f |
|---|---|---|---|---|---|---|---|---|---|---|---|
| GPT2-Tiny-MoE | 3.2M | 50.4M | OpenWebText | 12 | 4 | 256 | 256 | 512 | 1 | 2 | 1.0 |
| BERT-Large-MoE | 25.2M | 806.5M | wikitext-103 | 24 | 4 | 512 | 512 | 1024 | 2 | 1 | 1.0 |
| LLaMA2-MoE | 134.2M | 4297.6M | wikitext-103 | 32 | 4 | 512 | 1024 | 4096 | 1 | 1 | 1.0 |
| LLaMA2-MoE-L | 268.4M | 8595.2M | wikitext-103 | 64 | 4 | 512 | 1024 | 4096 | 1 | 1 | 1.0 |
| DeepSeek-V2-S | 419.6M | 2014.1M | OpenWebText | 4 | 4 | 256 | 5120 | 1536 | 2 | 8 | 1.0 |
| DeepSeek-V2-M | 734.3M | 3524.7M | OpenWebText | 7 | 4 | 256 | 5120 | 1536 | 2 | 1 | 1.0 |

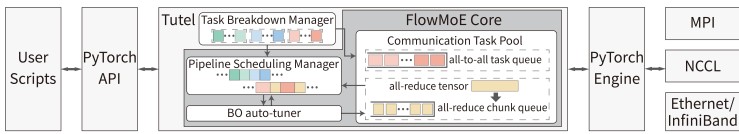

Figure 5: Overview of FlowMoE architecture (dark gray parts are new).

## 4.2 Algorithm Design

Algorithm 1 formally provides the pipeline scheduling training process of $R$-degree computing and communication tasks using FlowMoE. Moreover, Algorithm 2 provides the communication pool management, including the splitting of all-reduce tensors and the priority scheduling of communication tasks. A detailed description of the two algorithms is provided in Appendix E.

Compared to the vanilla expert parallelism [19], the overhead of pipeline scheduling in FlowMoE mainly comes from two parts: the partition of all-reduce tensors and the objective function fitting during BO. First, when the structure of the gating function is a typical linear layer of $M \times E$, the total number of parameters in the MHA layer and gating function is $4M^2 + M \times E$. According to Algorithm 2, the computational complexity of the scheduling algorithm of FlowMoE in each iteration is $\mathcal{O}(L \times \frac{4M^2 + M \times E}{S_p})$, which is acceptable because it is less than 1% of one iteration time in practical experiments, and the time of all-reduce tensor partitioning can be further overlapped with the time of some computing/communication tasks (we discuss its scalability in Appendix E).

Second, BO utilizes the first several iterations of the training process to automatically tune $S_p$ to a near-optimal value. In our experiments, BO samples 8 values of $S_p$ and record the iteration time corresponding to each value by averaging 10 iterations. The computational overhead introduced by BO is negligible compared to the total training time, as quantified in Appendix D. Additionally, if the hardware environment changes, BO will be re-executed to tune $S_p$, which is discussed in Appendix K. These details are omitted from the presentation of Algorithm 1 to focus more on the pipeline scheduling process of FlowMoE.

## 4.3 System Implementation

We deploy FlowMoE in PyTorch with its API, which usually supports class inheritance with flexible Python language and has a high generality to be compatible with different optimization frameworks and communication stacks. In particular, we implement FlowMoE atop Tutel [12], a highly optimized MoE acceleration library that is deeply integrated into PyTorch and supports asynchronous execution of communication and computing tasks. Tutel has also been used as a default MoE training module by DeepSpeed [38]. Fig. 5 shows the overview of FlowMoE architecture (the left side is closer to the user level), where its detailed description and technical implementations can be found in Appendix F.

# 5 Evaluation

## 5.1 Experimental Settings

**Testbed setup.** We use two clusters. (1) *Cluster 1* consists of 2 nodes connected with 100Gb/s bandwidth. Each node is equipped with 8 NVIDIA RTX3090 GPUs (24 GB of memory per GPU) connected with PCIe3.0x16. The CPU is Intel Xeon(R) Gold 6248R. (2) *Cluster 2* consists of 4 nodes connected with 10Gb/s bandwidth. Each node is equipped with 2 NVIDIA RTX2080Ti GPUs (12 GB of memory per GPU) connected with PCIe switches. The CPU is Intel Xeon(R) Gold 5118.

**Models with MoE Layers.** (1) *Customized MoE layers:* We cover typical configurations of MoE layers by choosing combinations of input parameters, where $B \in \{2, 4, 8\}$, $f \in \{1.0, 1.1, 1.2\}$, $N \in \{512, 1024, 2048\}$, $M \in \{512, 1024, 2048, 4096, 8192\}$ and $H \in \{512, 1024, 2048, 4096, 8192\}$.

Table 3: Comparison of average per-iteration time in milliseconds. S1, S2, S3, S4 and S5 are the speedups of FlowMoE over ScheMoE, FSMoE, Tutel, FasterMoE, and vanillaEP, respectively.

| # of GPUs | Model | Time (ms) | | | | | | $S_5$ | $S_4$ | $S_3$ | $S_2$ | $S_1$ |
|---|---|---|---|---|---|---|---|---|---|---|---|---|
| | | vanillaEP | FasterMoE | Tutel | FSMoE | ScheMoE | FlowMoE | | | | | |
| 4 | GPT2-Tiny-MoE | 104.2 | 90.1 | 85.6 | 82.8 | 80.8 | 66.1 | 1.58× | 1.36× | 1.29× | 1.25× | 1.22× |
| | BERT-Large-MoE | 373.1 | 300.9 | 314.6 | 278.1 | 273.6 | 239.5 | 1.56× | 1.26× | 1.31× | 1.16× | 1.14× |
| | LLaMA2-MoE | 1262.3 | 1187.5 | 1029.5 | 928.1 | 957.7 | 763.9 | 1.65× | 1.55× | 1.34× | 1.21× | 1.25× |
| | DeepSeek-V2-S | 2789.7 | 2261.4 | 2276.5 | 2096.3 | 2166.1 | 1740.8 | 1.60× | 1.29× | 1.30× | 1.20× | 1.24× |
| 8 | GPT2-Tiny-MoE | 125.1 | 119.3 | 116.8 | 98.8 | 107.5 | 87.6 | 1.43× | 1.36× | 1.33× | 1.13× | 1.23× |
| | BERT-Large-MoE | 428.8 | 354.4 | 377.5 | 345.1 | 331.2 | 283.2 | 1.51× | 1.25× | 1.33× | 1.22× | 1.17× |
| | LLaMA2-MoE | 1563.3 | 1427.7 | 1187.9 | 1110.4 | 1089.9 | 906.0 | 1.73× | 1.57× | 1.31× | 1.23× | 1.20× |
| | DeepSeek-V2-S | 4037.8 | 3370.5 | 3351.7 | 2985.4 | 3138.3 | 2384.9 | 1.69× | 1.41× | 1.39× | 1.25× | 1.31× |
| 16 | GPT2-Tiny-MoE | 169.5 | 135.3 | 129.3 | 114.8 | 116.4 | 95.6 | 1.77× | 1.41× | 1.35× | 1.20× | 1.22× |
| | BERT-Large-MoE | 537.8 | 490.8 | 501.1 | 421.9 | 405.6 | 351.9 | 1.53× | 1.39× | 1.42× | 1.19× | 1.15× |
| | LLaMA2-MoE | 1987.7 | 1759.1 | 1534.1 | 1292.6 | 1374.3 | 1124.0 | 1.76× | 1.57× | 1.36× | 1.15× | 1.22× |
| | DeepSeek-V2-S | 5843.3 | 4562.5 | 4481.4 | 3895.6 | 4093.7 | 3205.3 | 1.82× | 1.42× | 1.39× | 1.22× | 1.28× |

Table 4: Average per-iteration time with different pipelining degrees on DeepSeek-V2-S. S1 and S2 are the speedups of FlowMoE over Tutel and ScheMoE, respectively.

| $R$ | Time (ms) | | | $S_2$ | $S_1$ |
|---|---|---|---|---|---|
| | Tutel | ScheMoE | FlowMoE | | |
| 2 | 4481.4 | 4093.7 | 3205.3 | 1.39× | 1.28× |
| 4 | 4628.2 | 4164.0 | 3113.8 | 1.48× | 1.33× |
| 8 | 4588.9 | 4308.7 | 3295.9 | 1.39× | 1.30× |

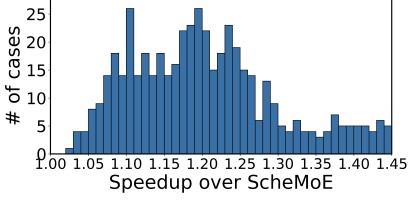
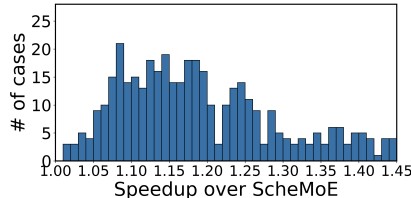

(a) Speedup on Cluster 1.     (b) Speedup on Cluster 2.

Figure 6: Statistic of the speedup over ScheMoE.

We set the number of experts equal to the number of GPUs (i.e., $E = P$) and $k = 2$. (2) *Real-world MoE models:* We also choose four popular models, i.e., GPT2-Tiny-MoE and DeepSeek-V2-S for the language modeling task on the OpenWebText dataset [39], and BERT-Large-MoE and LLaMA2-MoE for the text generation task on the wikitext-103 dataset [40]. We replace all feed-forward layers in GPT2-Tiny [41], BERT-Large [42], and LLaMA2 [43] with MoE layers to construct MoE models. Meanwhile, due to restricted memory, DeepSeek-V2-S only keeps four transformer blocks with MoE layers [44]. The detailed configuration is shown in Table 2. All parameters and gradients of MoE models are stored as 32-bit single precision floating point numbers.

**Baselines.** We compare FlowMoE with PyTorch-based vanilla expert parallelism (vanillaEP) [19] and existing state-of-the-art MoE frameworks, including ScheMoE [10], FSMoE [24], Tutel [12] and FasterMoE [11], with a pipelining degree of $R = 2$ unless specified otherwise. We use the official implementations of these frameworks. We focus on per-iteration training time, energy consumption and memory usage as performance metrics. All the reported numbers are averaged over 1000 iterations.

### 5.2 Experimental Results

**End-to-end Time on Real-world MoE Models.** We measure the average iteration time for end-to-end training of the models listed in Table 2 on Cluster 1. The experimental results are shown in Table 3. The results show that FlowMoE obtains the best scalability and the shortest per-iteration time on different sizes of clusters and models. Specifically, FlowMoE is 14%-31%, 13%-25%, 29%-42%,

Table 5: Per-iteration time of the MoE layer under different components. The reported speedup values are based on vanillaEP as the baseline. 'w/ Pipe-MoE' indicates pipelining expert computing and A2A communication. 'w/ Pipe-AT' indicates pipelining MHA layer computing and gating. 'w/ Pipe-AR' indicates pipelining all-reduce communication.

| Name | w/ Pipe-MoE | w/ Pipe-AT | w/ Pipe-AR | Time (ms) | Speedup |
|---|---|---|---|---|---|
| vanillaEP | × | × | × | 1630.8 | 1.0× |
| Tutel | √ | × | × | 1115.2 | 1.46× |
| FlowMoE-AT | √ | √ | × | 1012.6 | 1.61× |
| FlowMoE-AR | √ | × | √ (w/o BO) | 971.5 | 1.68× |
| FlowMoE-AR(BO) | √ | × | √ (w/ BO) | 895.3 | 1.82× |
| FlowMoE | √ | √ | √ | 796.1 | 2.05× |

26%-57% and 43%-82% faster than ScheMoE, FSMoE, Tutel, FasterMoE and vanillaEP, respectively. We also execute stress tests on different training frameworks across two scaled-up MoE models (LLaMA2-MoE-L and DeepSeek-V2-M) in Appendix G. Moreover, in most cases, Tutel is faster than FasterMoE due to its dedicated optimization for MoE training. Both FlowMoE and ScheMoE are built atop Tutel and outperform Tutel. ScheMoE and FSMoE further reduce training time by optimizing scheduling within the MoE layer and pipelining intra- and inter-node communication tasks, and this strategy can also be integrated into FlowMoE. In contrast, our proposed FlowMoE maximally overlaps communication with computing by pipelining all types of tasks. In bandwidth-limited scenarios, when the number of GPUs scales up, pipeline scheduling benefits less and longer communication time dominates the whole training process, thus the advantages of FlowMoE and other state-of-the-art MoE frameworks are not obvious. Moreover, FlowMoE essentially only changes the scheduling order of different tasks and does not affect the model convergence. We verify it through theoretical analysis and practical convergence experiments in Appendix H. Furthermore, we discuss the performance lower bound of FlowMoE in Appendix I and report the GPU SM utilization when training with FlowMoE in detail in Appendix J. We evaluate the sensitivity of FlowMoE to $S_p$ selection and BO hyperparameters, while measure the computational overhead of BO in Appendix D. We also discuss the robustness of FlowMoE to heterogeneous clusters, dynamic hardware environments, and node dropouts in Appendix K.

**Pipelining degree v.s. Training Speed.** Table 4 presents the averaged iteration time with different $R$ values on Cluster 1 with 16 GPUs. FlowMoE consistently outperforms ScheMoE and Tutel. Prior work [21] has provided an effective method to select the optimal $R$ by balancing overlapping degree and startup overhead of the tasks, which can be directly applied to FlowMoE. Orthogonal to the methods in choosing the best $R$, our scheduling framework focuses on pipelining all computing and communication tasks efficiently for any given $R \geq 2$.

**Speedup on Customized MoE Layers.** We construct the MoE layer using a combination of input parameters from customized MoE layer. We collect 490 valid cases from Cluster 1 with 16 GPUs and 393 cases from Cluster 2 with 8 GPUs (excluding out-of-memory cases) successfully measured on ScheMoE and FlowMoE, respectively. FlowMoE is always faster than ScheMoE in all valid cases. The speedup statistic of FlowMoE over ScheMoE is shown in Fig. 6. On average, FlowMoE achieves 26% improvement over ScheMoE.

**Ablation Study.** We conduct the ablation study on a customized MoE layer with $B = 4$, $f = 1.2$, $N = 512$, $M = 8192$, and $H = 8192$. The per-iteration time on Cluster 1 with different components using 16 GPUs is shown in Table 5. It is seen that FlowMoE-AT improve the time performance by 10.3% over Tutel by pipelining the MHA layer computing and gating. Further, FlowMoE-AR(BO) utilizes the priority scheduling mechanism of communication tasks based on all-reduce tensor chunks to pipeline the all-reduce communication and improve the time performance by another 24.6% over Tutel. In particular, our designed BO finds the near-optimal $S_p$ and FlowMoE-AR(BO) obtains 8.3% improvement in training time compared to FlowMoE-AR (where we set $S_p$ = 1MB). Putting our two optimizations together, FlowMoE runs 1.4× and 2.05× faster than Tutel and vanillaEP, respectively.

**Energy Consumption.** We use the NVIDIA SMI tool to sample the real-time power of each GPU in the cluster every 5ms, then integrate these sampled data over time to calculate the energy consumption of each GPU during the training process. We sum the energy consumption of all GPUs to obtain the total energy consumption, then divide it by the number of GPUs and the number of iterations to obtain the average per-worker energy consumption as shown in Table 6. The results indicate that a

Table 6: Averaged per-worker energy consumption and memory usage in one iteration.

| Model | vanillaEP | FasterMoE | Tutel | ScheMoE | FlowMoE |
|---|---|---|---|---|---|
| GPT2-Tiny-MoE | 1.7J/2.45GB | 1.5J/2.67GB | 1.3J/2.52GB | 1.2J/2.46GB | 1.0J/2.42GB |
| BERT-Large-MoE | 5.5J/4.19GB | 5.9J/4.97GB | 5.1J/4.13GB | 4.1J/4.16GB | 3.7J/3.89GB |
| LLaMA2-MoE | 20.2J/12.43GB | 19.9J/16.11GB | 15.6J/11.59GB | 14.0J/11.81GB | 12.1J/11.01GB |
| DeepSeek-V2-S | 59.5J/19.42GB | 54.9J/20.93GB | 45.6J/19.34GB | 41.7J/18.92GB | 34.9J/17.57GB |

higher overlapping degree of computing and communication tasks leads to higher computing and communication resource utilization, thus exhibiting lower energy consumption. Overall, FlowMoE saves 10%-16%, 22%-27%, 33%-39% and 33%-41% energy consumption compared to ScheMoE, Tutel, FasterMoE and vanillaEP, respectively. It is worth noting that the power reported by the NVIDIA SMI tool includes both communication and computing costs. According to the official documentation of NVIDIA SMI [45], the energy consumption measured by the NVIDIA SMI tool covers the entire GPU card's power consumption, where communication power includes (1) PCIe communication costs, (2) NVLink/NVSwitch communication costs, and (3) collective communication costs (e.g., NCCL All-Reduce and A2A). Therefore, our measurement results effectively reflect the total cluster-wide energy usage, compatibly providing a fair comparison with the baselines.

**Memory Usage.** Table 6 also demonstrates average memory usage per worker by monitoring all GPUs every 1s on Cluster 1 with 16 GPUs using the NVIDIA SMI tool. FlowMoE has the lowest memory usage because it performs all-reduce communication tasks in time to reduce gradient caching. Specifically, FlowMoE saves up to 7%, 9%, 32% and 11% of memory compared to ScheMoE, Tutel, FasterMoE and vanillaEP, respectively. In addition, the memory usage of ScheMoE and Tutel is similar to that of vanillaEP since there is no optimization in the memory. FasterMoE needs to replicate expert parameters among workers in its load balancing mechanism and consumes more GPU memory.

## 6 Related Work

In addition to the frameworks mentioned in Section 2 that focus on computing-communication pipelines in distributed training, we also discuss some other works. First, general pipeline schedulers such as PipeDream [26] and Gpipe [25] enable pipelining by splitting the model across multiple GPUs, requiring communication between GPUs to exchange activation values or gradients across split layers. In contrast, FlowMoE implements pipelining of computing-communication tasks, accelerating model training by minimizing communication overhead. Second, several orthogonal MoE optimization techniques can be combined with FlowMoE. (1) NetMoE [46] alleviates communication bottlenecks and load imbalance by reducing cross-node token routing and dynamically adjusting expert placement. (2) Lancet [47] determines task scheduling order by constructing a computing-communication directed acyclic graph (DAG), without considering microbatch-level partitioning of the same task. (3) Punniyamurthy et al.'s work [48] focuses on fusing computing-communication operations and reducing kernel-level startup overhead. However, it does not consider scheduling relationships among all major MoE tasks or how to maximize computing-communication overlap.

## 7 Conclusion

In this paper, we propose a scalable and mathematically proven pipeline scheduling framework called FlowMoE to accelerate the training of MoE models. FlowMoE addresses the unified scheduling across all major MoE-related tasks and enables the optimal coexistence of heterogeneous communication tasks (A2A and all-reduce communications) in MoE training. This substantially advances distributed solutions beyond a simple extension of traditional pipeline methods. We implement the FlowMoE framework on Pytorch. We conduct extensive experiments with 675 typical MoE layers and four real-world NLP models across two GPU clusters. Experimental results show that FlowMoE outperforms state-of-the-art MoE training frameworks including ScheMoE, FSMoE, Tutel, and FasterMoE by 13%-57% in training time, 10%-39% in energy consumption, and 7%-32% in memory usage. Furthermore, FlowMoE can be combined with many orthogonal optimization works, promoting distributed pipeline optimization to a new stage.

## 8 Acknowledgements

This work was supported in part by the National Key Research and Development Project under Grant 2022YFB2901604, and in part by Zhejiang Provincial Natural Science Foundation of China under Grant LZ22F010008.

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

# Appendix

## FlowMoE: A Scalable Pipeline Scheduling Framework for Distributed Mixture-of-Experts Training

Table A.1: Frequently used notations

| Name | Description |
|------|-------------|
| $P$ | Number of workers (or GPUs) in the cluster. |
| $L$ | Number of transformer blocks in a MoE model. |
| $B$ | Number of samples per GPU (or mini-batch size) in one iteration. |
| $N$ | Number of tokens per sample. |
| $M$ | Embedding size of a token. |
| $E$ | Total number of experts per MoE layer. |
| $H$ | Hidden size of the feed-forward layer in experts. |
| $k$ | Top-$k$ experts should be selected for each token. |
| $f$ | Capability factor that determines the maximum number of tokens assigned to each expert. |
| $R$ | Pipelining degree. |
| $S_p$ | Partition size of the all-reduce tensor block. |
| $AT_r^{(l)}$ | The $r$th MHA layer computing subtask of the $l$th transformer block (including the $r$th gating subtask) |
| $E_r^{(l)}$ | The $r$th expert computing subtask of the $l$th transformer block. |
| $D_r^{(l)}$ | The $r$th dispatch subtask (A2A communication) of the $l$th transformer block. |
| $C_r^{(l)}$ | The $r$th combining subtask (A2A communication) of the $l$th transformer block. |
| $AR^{(l)}$ | The all-reduce communication task of the $l$th transformer block during the backward propagation. |
| $\tau_f(\cdot)/\tau_b(\cdot)$ | The beginning execution timestamp of a task during the feed-forward computing/backward propagation. |
| $t_f(\cdot)/t_b(\cdot)$ | The elapsed time of a task during the feed-forward computing/backward propagation. |

## A  Main differences between FlowMoE and the key literature

Table A.2: Main differences between FlowMoE and the key literature on pipeline scheduling for distributed MoE training, where BO represents the Bayesian optimization.

| Scheduling framework | vanillaEP[19] | FasterMoE[11] | Tutel[12] | ScheMoE[10] | FlowMoE |
|----------------------|---------------|---------------|-----------|-------------|---------|
| A2A pipelining | × | √ | √ | √ | √ |
| Expert computing pipelining | × | √ | √ | √ | √ |
| MHA + gating pipelining | × | × | × | × | √ |
| All-reduce pipelining | × | × | × | × | √ |
| Auto-tuning | × | × | × | × | √ (BO) |
| Robustness of dynamic hardware environment | Weak | Weak | Weak | Weak | Strong |
| Speedup (Homogeneous 16-GPU cluster) | 1.0× | 1.28× | 1.31× | 1.45× | 1.82× |
| Speedup (Heterogeneous 16-GPU cluster) | 1.0× | 1.31× | 1.36× | 1.44× | 1.74× |

## B  Proof of Theorem 1

According to Eqs. 4 and 5, since the task scheduling timeline in each transformer block is the same and the all-reduce communication time of each transformer block is also the same, we only need to

prove that $T_b$ when inserting $AR^{(l+1)}$ between any two A2A communication tasks of transformer block $l$ will be less than or equal to $T_b^*$.

We discuss the relationship between $T_b$ and $T_b^*$ under the following four tentative scheduling orders ($1 < l < L$, $1 \le r < R$):

Scheduling order 1: $\tau_b(C_{r+1}^{(l)}) \le \tau_b(AR^{(l+1)}) \le \tau_b(C_r^{(l)})$.

Scheduling order 2: $\tau_b(C_1^{(l)}) \le \tau_b(AR^{(l+1)}) \le \tau_b(D_R^{(l)})$.

Scheduling order 3: $\tau_b(D_{r+1}^{(l)}) \le \tau_b(AR^{(l+1)}) \le \tau_b(D_r^{(l)})$.

Scheduling order 4: $\tau_b(D_1^{(l)}) \le \tau_b(AR^{(l+1)}) \le \tau_b(C_R^{(l-1)})$.

If Scheduling order 1 holds, according to Eq. 6b, we have

$$\tau_b(E_r^{(l)}) = \max\{\tau_b(C_r^{(l)}) + t_b(C_r^{(l)}) + t_b(AR^{(l+1)}), \tau_b(E_{(r+1)}^{(l)}) + t_b(E_{(r+1)}^{(l)})\}. \tag{A.1}$$

And if $AR^{(l+1)}$ is performed at the end of backward propagation, according to Eq. 6b, we have

$$\tau_b^*(E_r^{(l)}) = \max\{\tau_b^*(C_r^{(l)}) + t_b(C_r^{(l)}), \tau_b^*(E_{(r+1)}^{(l)}) + t_b(E_{(r+1)}^{(l)})\}. \tag{A.2}$$

Since $\tau_b^*(C_r^{(l)}) = \tau_b(C_r^{(l)})$, according to Eqs. A.1 and A.2, we can derive

$$\tau_b(E_r^{(l)}) \le \tau_b^*(E_r^{(l)}) + t_b(AR^{(l+1)}). \tag{A.3}$$

Meanwhile, according Eqs. 4, 5 and 6a-6e, we have

$$\tau_b(AR^{(1)}) + t_b(AR^{(1)}) - \tau_b(E_r^{(l)}) + t_b(AR^{(l+1)}) = \tau_b^*(AR^{(1)}) + t_b(AR^{(1)}) - \tau_b^*(E_r^{(l)}). \tag{A.4}$$

In other words, between task $AR^{(1)}$ and task $E_r^{(l)}$, the timeline based on centralized scheduling of all-reduce communication tasks has one more $AR^{(l+1)}$ than the timeline based on Scheduling order 1. Then, adding the left and right terms of Eqs. A.3 and A.4, respectively, we have

$$\tau_b(AR^{(1)}) + t_b(AR^{(1)}) \le \tau_b^*(AR^{(1)}) + t_b(AR^{(1)}). \tag{A.5}$$

Moreover, since $\tau_b(C_R^{(L)}) = \tau_b^*(C_R^{(L)})$, according to Eq. 6, we have $T_b \le T_b^*$. An easy-to-understand demonstration is presented in Fig. A.1.

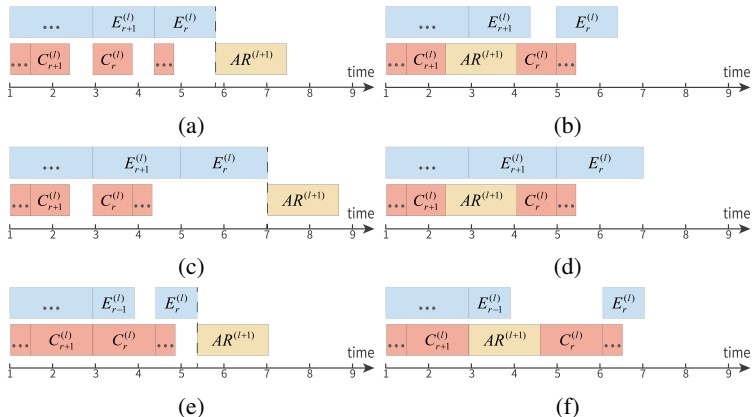

Figure A.1: Demonstration of one example where (a), (c) and (e) represent the three timelines when using centralized scheduling of all-reduce communication tasks (the dotted lines indicate that the tasks before and after do not have direct dependency), and (b), (d) and (f) represent the three timelines when using Scheduling order 1. Comparing (a) with (b), (c) with (d) and (e) with (f), it can be observed that $T_b \le T_b^*$.

Similarly, we can obtain the same conclusion when using the tentative scheduling orders 2,3 and 4. Therefore, $T_b$ will be less than or equal to $T_b^*$ when inserting all-reduce communication task of one transformer block between any two A2A communication tasks, and the proof is complete.

# C Proof of Theorem 2

When the scheduling order satisfies Eqs. 4 and 5 during backward propagation, the idle time of the communication resources (i.e., the gaps between the A2A communication tasks) is fixed if not considering the all-reduce communication tasks. Obviously, the approach to minimize the time of one iteration is to maximize the occupation of the idle time of the communication resources by the communication tasks of the all-reduce tensor chunks. Furthermore, if the communication of the all-reduce tensor chunks does not introduce extra startup overhead, the total time of the all-reduce communication tasks will remain unchanged for different $S_p$. When $S_p$ is larger, A2A communication tasks may not be executed in time due to uncompleted communication tasks of the all-reduce tensor chunks, which will cause the timeline of the computing tasks to shift back according to Eqs. 6b and 6d and part of the communication tasks of the all-reduce tensor chunks do not occupy the free time of the communication resource (see Figs. A.1b and A.1f). On the contrary, when $S_p \rightarrow 0$, the communication tasks of the all-reduce tensor chunks will not affect the timeline of the A2A communication tasks and thus will not change the timeline of the computing tasks due to the higher priority of the A2A communication tasks. Meanwhile, it will maximize the free time of the communication resources. Thereby, the proof is complete.

# D Design and Performance Evaluation of BO

## D.1 Parameter Settings and Advantage Analysis

We choose BO to obtain a near-optimal $S_p$ benefiting from the following three points:

- First, BO is not limited by the expression of the objective function (we use $\mathcal{F}(S_p)$ as the objective function) and depends only on the sampling values obtained (i.e., $\hat{\mathcal{F}}(S_p^1), \hat{\mathcal{F}}(S_p^2), ..., \hat{\mathcal{F}}(S_p^n)$). We use Gaussian process regression with the Matern kernel to predict the value of the objective function as it is commonly used as a good surrogate model for BO [49]. A 95% confidence interval is associated with $\mathcal{F}(S_p)$, which is considered the most likely region for the value of $\mathcal{F}(S_p)$.

- Second, BO typically requires only a limited number of trials to find high-quality solutions, resulting in low search overhead. To minimize the number of trials, it selects the next configuration $S_p$ by maximizing an acquisition function [49]. In FlowMoE, we adopt the Expected Improvement (EI) acquisition function to choose the $S_p$ that can maximize training speedup improvement compared to the current best result.

- Third, BO avoids the search process from falling into a local optimum by tuning the hyperparameter Expected Improvement (EI). Specifically, a smaller EI favors exploitation, i.e., collecting more points near the peak, while a larger EI tends to exploration, i.e., collecting more scattered points in the range [49]. In FlowMoE, we choose EI=0.1 to prefer $S_p$ exploration, which is a commonly adopted value [31, 50]. Meanwhile, the search space of $S_p$ in BO can be set as (0MB, the maximum value of the tensor to be communicated in each transformer block] and the required one initial sample value is randomly generated, which can effectively cover the optimal $S_p$ for different hardware clusters and MoE models, e.g., from 0MB-10MB in Fig. 4.

## D.2 Importance Analysis of BO

BO auto-tuning is an indispensable part of FlowMoE and is absolutely necessary. The performance of FlowMoE depends on the introduced BO process. Specifically, an oversized all-reduce tensor chunk will affect the prioritization of A2A communication tasks and may prevent them from being started in time. In contrast, an undersized all-reduce tensor chunk will result in excessive startup overhead. Therefore, neither an oversized nor undersized all-reduce tensor chunk can achieve the shortest per-iteration time, and there must exist a unique optimal solution that maximizes training speed. The BO optimizer balances these two competing effects by sampling and learning from actual iterations, efficiently finding an all-reduce tensor chunk size that maximizes overlap without incurring excessive overhead. This tuning is crucial because the optimal trade-off point varies across models, GPU interconnect topologies, and MoE configurations.

### D.3 Performance Evaluation of BO

Table A.3: Comparison of average per-iteration time in milliseconds when using different approaches to tune the partitioning size $S_p$.

| Model | Time (ms) | | |
|---|---|---|---|
| | BO | Grid Search | Random Number Generation |
| GPT2-Tiny-MoE | 95.6 | 101.3 | 109.3 |
| BERT-Large-MoE | 351.9 | 373.80 | 388.96 |
| LLaMA2-MoE | 1124.0 | 1208.23 | 1250.09 |
| DeepSeek-v2-S | 3205.3 | 3498.8 | 3902.75 |

We compare the performance of BO with the other two methods including grid search and random number generation on tuning $S_p$. In our experiments, we set the other two methods to use the same search space as BO. For grid search, the search space is divided into 8 equal parts to construct 8 discrete sampling points. Similarly, the grid search utilized the first 80 iterations of the training process to determine $S_p$. Specifically, each iteration time corresponding to each sample point is obtained by averaging 10 iterations, and the sample point corresponding to the smallest per-iteration time is used as the value of $S_p$ for subsequent training. For random number generation, we randomly selected a number from the search space as the value of $S_p$ in each iteration. Table A.3 shows the comparison of average iteration times when using different methods of tuning $S_p$ in FlowMoE on Cluster 1 with 16 GPUs. It can be observed that using BO to tune $S_p$ obtained the shortest per-iteration time among the three methods. Although grid search obtains better performance than random number generation, its limited number of sampling points makes it difficult to cover the optimal solution especially when the search space is large, and over-increasing the number of sampling points brings higher search overhead and longer search process. On the contrary, BO can obtain a near-optimal $S_p$ with very little overhead, and therefore we choose it.

Table A.4: Comparison of average per-iteration time in milliseconds when using FlowMoE with BO auto-tuning or different fixed partition sizes.

| Model | Time (ms) | | | | | |
|---|---|---|---|---|---|---|
| | BO | $S_p$=0.5MB | $S_p$=1MB | $S_p$=2MB | $S_p$=4MB | $S_p$=8MB |
| GPT2-Tiny-MoE | 95.6 | 130.1 | 115.9 | 104.7 | 109.7 | 122.3 |
| BERT-Large-MoE | 351.9 | 388.9 | 378.6 | 362.2 | 386.3 | 395.1 |
| LLaMA2-MoE | 1124.0 | 1213.8 | 1167.9 | 1185.9 | 1211.8 | 1240.5 |
| DeepSeek-v2-S | 3205.3 | 4438.5 | 3948.9 | 3654.7 | 3493.9 | 3740.2 |

**BO Auto-tuning v.s. Different Fixed Partition Sizes.** In addition, we count the average per-iteration time when using FlowMoE with BO auto-tuning or different fixed partition sizes for training four MoE models (see Table 2 for their detailed configurations) in Cluster 1 with 16 GPUs to validate the sensitivity of FlowMoE to $S_p$. As shown in Table A.4, different partition sizes greatly affect the training efficiency and BO auto-tuning is essential to maximize the performance of FlowMoE.

Table A.5: Comparison of average per-iteration time when training BERT-Large-MoE with different BO parameter configurations on Cluster 1, where the surrogate model uses Gaussian Process Regression (GPR) with different kernel functions.

| BO Hyperparameter | | Time (ms) |
|---|---|---|
| Acquisition Function | Surrogate Model | |
| Expected Improvement (EI=0.1) | GPR + Matern | 351.9 |
| Expected Improvement (EI=0.05) | GPR + Matern | 358.9 |
| Expected Improvement (EI=0.2) | GPR + Matern | 354.2 |
| Probability of Improvement | GPR + Matern | 355.1 |
| Lower Confidence Bound | GPR + Matern | 355.4 |
| Expected Improvement (EI=0.1) | GPR + RBF | 357.2 |
| Expected Improvement (EI=0.1) | GPR + Rational Quadratic | 360.2 |

**Sensitivity of BO hyperparameters.** In FlowMoE, the objective function for per-iteration time regarding the all-reduce tensor chunk size is a single-peaked, smooth objective function with a fixed search space. For this objective function, the process of using BO to find the optimal solution is insensitive to the two hyperparameters (acquisition function and surrogate model) of BO, and BO always converges and approaches the optimal solution. Table A.5 shows the comparison results of average per-iteration time when training BERT-Large-MoE with different BO parameter configurations on Cluster 1. The results indicate that although the performance of FlowMoE is sensitive to the BO process, BO hyperparameters have a minor impact on it, and different BO configurations lead to similar iteration time.

Table A.6: Percentage of the computational overhead of BO to the training time of first 1000 iterations.

| Model | GPT2-Tiny-MoE | BERT-Large-MoE | LLaMA2-MoE | DeepSeek-v2-S |
|---|---|---|---|---|
| Overhead | 3.22% | 1.38% | 0.43% | 0.16% |

**Computational Overhead of BO.** We also measure the percentage of the computational overhead of BO to the training time of first 1000 iterations when training four MoE models. The experimental results are illustrated in Table A.6. It can be observed that this overhead is negligible compared to the gains brought by BO (see Table 3), and it will be smaller when training the model to convergence, which involves tens of thousands of iterations. Thus, the auto-tuning process of BO is lightweight and practical.

# E    Algorithm Description and Scalability Analysis

Algorithm 1 describes the pipeline of $R$-degree computing and communication tasks during the training process. Line 1-4 initializes the necessary parameters and queues. Line 6-12 performs the feed-forward computing of each iteration according to Eqs. 2 and 3. Meanwhile Line 13-21 performs the backward propagation of each iteration according to Eqs. 4 and 5. Algorithm 2 shows the management in the communication pool. The PARTITION procedure is responsible for splitting the tensor of the MHA layer and gating function during the backward propagation process and placing it into the queue of the all-reduce communication tasks.The COMMPOOLMANAGER procedure performs the two types of communication tasks according to the defined priorities, where Line 9-11 prioritizes the execution of the A2A communication tasks and queues the data obtained after the communication, while Line 12-13 executes the communication tasks of the all-reduce tensor chunks when there are no A2A communication tasks.

To analyze the scalability of FlowMoE's scheduling algorithm, we examine the computational complexity of each iteration of the MoE model in distributed training. We assume that each GPU processes the same number of tokens and do not consider the non-overlapped time of A2A communication and all-reduce communication in each iteration, then the time complexity of one iteration is $\mathcal{O}\left(L \times \left[BNM^2 + B^2N^2M + BNME + 2BNMH\right]\right)$, where $\mathcal{O}(BNM^2 + B^2N^2M)$ denotes the linear mapping of the Q,K,V and linear transformation matrix in the MHA layer and the attention score computation, $\mathcal{O}(BNME)$ denotes the gate function computing with the structure of $M \times E$, and $\mathcal{O}(2BNMH)$ denotes the expert computing on each GPU. Then, when the complexity of the MoE model scales up, the increase in the complexity of FlowMoE's scheduling algorithm is significantly smaller than the increase in the complexity of one iteration time. Meanwhile, when the number of GPUs $P$ in the cluster increases, the complexity of the scheduling algorithm of FlowMoE is not affected, but the complexity of one iteration time may increase due to the longer time of the communication tasks. In summary, the ratio of the time overhead of FlowMoE's scheduling algorithm to one iteration time will decrease when the model complexity and the number of GPUs increase. Therefore, FlowMoE's scheduling algorithm has good scalability.

# F    System Description

From the closest to user level to the lowest level, distributed training frameworks and communication stacks usually include: User Script level, PyTorch frontend with high-level APIs, PyTorch Engine, message-level communication library. To ensure the generality of FlowMoE, we can not modify user scripts and framework engines heavily. However, due to the diversity of MPI (e.g., OpenMPI [51],

Intel MPI [52]) and network interfaces (e.g., Ethernet, Infiniband) in the communication library level, it is almost impossible to make one piece of code work in all communication libraries. Therefore, FlowMoE is deployed at the PyTorch API level, where its four modules is elaborated as follow:

- Task Breakdown Manager is responsible for splitting the dataset of each batch size according to the degree $R$ and requesting different communication/computing subtasks based on the tensor transferred between different computing and communication tasks. Task Breakdown Manager has been improved based on Tutel.

- BO autotuner is responsible for searching the near-optimal partition size and guiding the partitioning of the all-reduce tensor.

- Communication Task Pool maintains queues of A2A communication tasks and all-reduce chunk communication tasks. On the one hand, it receives A2A communication tasks requested from the Task Breakdown Manager and queues them according to the request order. On the other hand, it splits the all-reduce tensor generated in the backward propagation and queues the all-reduce chunks.

- Pipeline Scheduling Manager submits the requested multiple computing tasks and communication tasks in the queue to the PyTorch Engine and the communication library according to the defined scheduling order and priority.

In addition, the backward propagation of each transformer block is not intuitively visible in PyTorch, and the gradients of the model parameters can only be accessed after the backward propagation is fully completed. To obtain the gradient tensors of the MHA layer and gating function for each transformer block in time during backward propagation, we use *register_full_backward_hook* to access the gradients and split the gradient tensors into queues based on the partition size $S_p$. Meanwhile, we achieve the overlapping of computing and communication tasks by using multi-threading. Specifically, the main thread includes Task Breakdown Manager, BO autotuner and Pipeline Scheduling Manager and is responsible for scheduling all tasks. We also create one sub-thread in Communication Task Pool to schedule the communication tasks in the two queues based on the defined priorities, and a *threading.Lock* is used to maintain security for all threads.

## G  Stress Tests in two Scaled-up MoE Models

Table A.7: Comparison of average per-iteration time when training two scaled-up MoE models. S1, S2, and S3 are the speedups of FlowMoE over ScheMoE, Tutel and vanillaEP, respectively.

| # of GPUs | Model | Time (ms) | | | | $S_3$ | $S_2$ | $S_1$ |
| --- | --- | --- | --- | --- | --- | --- | --- | --- |
| | | vanillaEP | Tutel | ScheMoE | FlowMoE | | | |
| 4 | LLaMA2-MoE-L | 2405.1 | 1927.0 | 1806.1 | 1493.8 | 1.61× | 1.29× | 1.21× |
| | DeepSeek-V2-M | 535.3 | 468.4 | 432.2 | 352.2 | 1.52× | 1.33× | 1.23× |
| 8 | LLaMA2-MoE-L | 2989.1 | 2493.9 | 2297.9 | 1833.8 | 1.63× | 1.36× | 1.25× |
| | DeepSeek-V2-M | 944.6 | 773.4 | 723.6 | 552.4 | 1.71× | 1.40× | 1.31× |
| 16 | LLaMA2-MoE-L | OOM | OOM | OOM | OOM | / | / | / |
| | DeepSeek-V2-M | 1254.6 | 956.9 | 893.4 | 708.8 | 1.77× | 1.35× | 1.26× |

To stress-test FlowMoE's performance, we measured the training performance of different training frameworks on two scaled-up MoE models (LLaMA2-MoE-L and DeepSeek-V2-M), both of which are close to the memory upper limit of Cluster 1. Table A.7 shows the average per-iteration time using different frameworks on these two MoE models (FasterMoE is OOM in any cases). The results indicate that FlowMoE still achieved the best training performance among all baselines on larger models.

## H  Convergence Analysis and Experiments

We denote the samples processed in each pipeline as one microbatch, and the number of microbatches is equal to the pipelining degree $R$.

Specifically, during the backpropagation of the $l$th transformer block in one iteration, the gradients from all $AT_r^l$ and $E_r^l$ $(1 \le r \le R)$ after backpropagation are accumulated and summed. Once the gradient from $AT_1^l$ is also accumulated, the All-Reduce chunk communication task for the MHA and gate function in the $l$th transformer block is started. Similarly, when the gradient of $E_1^l$ is accumulated, the expert parameters are updated. This effectively prevents the parameters of MHA, the gate function, and the expert from being updated early, thereby avoiding gradient staleness. Additionally, to ensure that the accumulated gradients are equivalent with and without pipelining, we scale the loss calculated for each microbatch by $R$, i.e., $\frac{loss^{(r)}}{R}(1 \le r \le R)$, where $loss^{(r)}$ is the loss calculated using the samples from the $r$th microbatch, and the detailed theoretical derivation is as follows:

The loss and gradient when performing backpropagation using the entire mini-batch are expressed as follows:

$$loss = \frac{1}{B}\sum_{i=1}^{B}\ell(x_i,y_i), \nabla L_{full} = \nabla\left(\frac{1}{B}\sum_{i=1}^{B}\ell(x_i,y_i)\right), \tag{A.6}$$

which is currently used by all mainstream MoE training frameworks [12, 38, 28].

When dividing mini-batch into $R$ microbatches, the number of samples in each microbatch is $b = \frac{B}{R}$. The loss for each microbatch is:

$$loss^{(r)} = \frac{1}{b}\sum_{i=1}^{b}\ell(x_{r,i},y_{r,i}). \tag{A.7}$$

Then, the scaled loss for each microbatch is:

$$\tilde{loss}^{(r)} = \frac{1}{R}\cdot loss^{(r)} = \frac{1}{R}\cdot\frac{1}{b}\sum_{i=1}^{b}\ell(x_{r,i},y_{r,i}) = \frac{1}{B}\sum_{i=1}^{b}\ell(x_{r,i},y_{r,i}). \tag{A.8}$$

The cumulative loss of all microbatches is:

$$\sum_{r=1}^{R}\tilde{loss}^{(r)} = \sum_{r=1}^{R}\frac{1}{B}\sum_{i=1}^{b}\ell(x_{r,i},y_{r,i}) = \frac{1}{B}\sum_{r=1}^{R}\sum_{i=1}^{b}\ell(x_{r,i},y_{r,i}) = \frac{1}{B}\sum_{i=1}^{B}\ell(x_i,y_i). \tag{A.9}$$

According to Eqs. A.6 and A.8, we can obtain:

$$\nabla\left(\sum_{r=1}^{R}\tilde{loss}^{(r)}\right) = \nabla\left(\frac{1}{B}\sum_{i=1}^{B}\ell(x_i,y_i)\right) = \nabla L_{full}. \tag{A.10}$$

This analysis shows that the gradient update strategy is numerically equivalent with and without pipelining. The only difference between them is the optimization of task scheduling order during execution, which does not compromise model convergence during training.

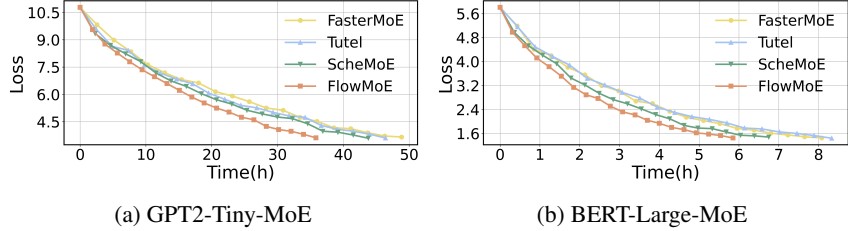

(a) GPT2-Tiny-MoE                                    (b) BERT-Large-MoE

Figure A.2: Loss with time for training GPT2-Tiny-MoE and BERT-Large-MoE.

Furthermore, we experimentally verify the convergence of FlowMoE by training GPT2-Tiny-MoE and BERT-Large-MoE on Cluster 1 with 16 GPUs. As shown in Fig. A.2, FlowMoE reaches the same loss as the baselines, and it takes much less time due to reduced per-iteration time.

# I  Performance Lower Bound Analysis

We analyze the performance lower bound of FlowMoE from the following three cases:

(1) **The communication task time is much longer than the computing task time.** When the communication time is very long, the time of the All-to-All (A2A) communication task alone completely covers the computing task time. In this case, the All-Reduce (AR) chunk based priority scheduling mechanism will fail because the AR chunk cannot be inserted into the A2A communication task. The performance of FlowMoE will be the same as ScheMoE, Tutel, and FasterMoE, but better than vanillaEP due to the hidden computing task time.

(2) **The computing task time is much longer than the communication task time.** When the computing time is very long, all the communication task time can be covered by the computing task time. In this case, FlowMoE will outperform ScheMoE, Tutel, and FasterMoE due to the hidden AR task time, and better than vanillaEP because it further hides the A2A task time.

(3) **The communication task time is comparable to the computing task time.** In this case, FlowMoE outperforms all baselines because it maximizes the overlap of multi-type tasks.

In summary, in all cases, the performance of FlowMoE is greater than or equal to that of ScheMoE, Tutel, and FasterMoE, and is always better than that of vanillaEP.

# J  GPU SM Utilization

Table A.8: Average GPU SM utilization with different microbatch sizes.

| Name | Model | $R$ | Average GPU SM Utilization |
|------|-------|-----|----------------------------|
| FlowMoE | GPT2-Tiny-MoE | 2 | 72.63% |
| FlowMoE | GPT2-Tiny-MoE | 4 | 48.43% |
| vanillaEP | GPT2-Tiny-MoE | / | 87.09% |
| FlowMoE | BERT-Large-MoE | 2 | 87.84% |
| FlowMoE | BERT-Large-MoE | 4 | 78.16% |
| vanillaEP | BERT-Large-MoE | / | 88.90% |
| FlowMoE | LLaMA2-MoE | 2 | 89.16% |
| FlowMoE | LLaMA2-MoE | 4 | 88.19% |
| vanillaEP | LLaMA2-MoE | / | 89.49% |
| FlowMoE | DeepSeek-V2-S | 2 | 89.27% |
| FlowMoE | DeepSeek-V2-S | 4 | 88.85% |
| vanillaEP | DeepSeek-V2-S | / | 90.77% |

First, we use the CUPTI tool to measure GPU SM utilization on Cluster 1 with 16 GPUs under different microbatch sizes. Specifically, we adjust the microbatch size by changing the pipelining degree $R$ (a larger $R$ results in a smaller microbatch). VanillaEP represents the original mini-batch without pipelining. The results are shown in Table A.8. We observe that smaller microbatch sizes may result in lower GPU SM utilization (e.g., when training GPT2-Tiny-MoE with $R = 4$ using FlowMoE), but in most cases, GPU SM utilization is nearly identical to that without pipelining (e.g., when training BERT-Large-MoE, LLaMA2-MoE and DeepSeek-V2-S using FlowMoE). This is because the actual tensors involved in GPU computation remain sufficiently large for larger MoE models, and GPU SM resources are still utilized efficiently. In other words, the microbatches introduced by FlowMoE's pipelining do not waste GPU SM resources for large MoE models.

Second, we measure GPU SM utilization with different batch sizes when training different MoE models using FlowMoE on Cluster 1 with 16 GPUs. As illustrated in Table A.9, smaller batch sizes are more likely to lead to lower GPU SM utilization, e.g., when training GPT2-Tiny-MoE or BERT-Large-MoE with a batch size of 2. Moreover, GPU SM utilization remained nearly unchanged when training LLaMA2-MoE or DeepSeek-V2-S. This is because these two MoE models have a larger number of parameters, which keeps the tensor dimensions involved in GPU computation sufficiently large, ensuring that SM resources are still efficiently utilized even with a lower batch size.

Table A.9: Average GPU SM utilization with different batch sizes when using FlowMoE.

| Model | Batch Size | Average GPU SM Utilization |
|---|---|---|
| GPT2-Tiny-MoE | 4 | 72.63% |
| GPT2-Tiny-MoE | 2 | 36.62% |
| BERT-Large-MoE | 4 | 87.84% |
| BERT-Large-MoE | 2 | 61.48% |
| LLaMA2-MoE | 4 | 89.16% |
| LLaMA2-MoE | 2 | 88.45% |
| DeepSeek-V2-S | 4 | 89.27% |
| DeepSeek-V2-S | 2 | 89.06% |

Table A.10: Parameter Configurations for BERT-Large-MoE-w

| MoE Model | # Params (MHA+Gating) | # Params (Experts) | Dataset | Configurations | | | | | |
|---|---|---|---|---|---|---|---|---|---|
| | | | | L | B | N | M | H | E/P k |
| BERT-Large-MoE-w | 25.2M | 3325.9M | wikitext-103 | 24 | 4 | 512 | 512 | 1024 | 8 1 |

Table A.11: The maximum and minimum GPU SM utilization for a large number of experts with different numbers of activated experts when using FlowMoE in Cluster 1 with 16 GPUs..

| Model | $f$ | Maximum GPU SM Utilization | Minimum GPU SM Utilization |
|---|---|---|---|
| BERT-Large-MoE-w | 1.0 | 89.20% | 87.81% |
| BERT-Large-MoE-w | 4.0 | 89.72% | 50.65% |
| BERT-Large-MoE-w | 8.0 | 90.30% | 31.60% |
| BERT-Large-MoE-w | 16.0 | 90.68% | 19.41% |

Third, we also evaluate the impact of the number of activated experts on GPU utilization when using FlowMoE on Cluster 1 with 16 GPUs for a large number of experts. Specifically, we construct BERT-Large-MoE-w by increasing the number of experts per GPU from 2 to 8 and adjusting the number of activated experts by modifying the capacity factor $f$, where a larger $f$ indicates more uneven token routing by the gate function and fewer activated experts since most tokens are routed to popular experts. The detailed configurations of BERT-Large-MoE-w are shown in Table A.10. We use the CUPTI tool and report the maximum and minimum GPU SM utilization in Table A.11. The results demonstrate that, with a large number of experts, the fewer the number of activated experts, the more unbalanced the computation load on the GPU and the greater the difference in SM utilization between GPUs.

# K   Robustness Analysis

We discuss the robustness of FlowMoE to heterogeneous clusters, dynamic hardware environments, and node dropouts.

## K.1   Robustness for Heterogeneous Clusters

Table A.12: Comparison of average iteration time in milliseconds when the GPUs have different computing power. S1, S2, S3 and S4 are the speedups of FlowMoE over vanillaEP, FasterMoE, Tutel and ScheMoE, respectively.

| Model | Time (ms) | | | | | $S_4$ | $S_3$ | $S_2$ | $S_1$ |
|---|---|---|---|---|---|---|---|---|---|
| | vanillaEP | FasterMoE | Tutel | ScheMoE | FlowMoE | | | | |
| GPT2-Tiny-MoE | 235.8 | 201.6 | 189.1 | 178.2 | 153.3 | 1.54× | 1.32× | 1.23× | 1.16× |
| BERT-Large-MoE | 657.7 | 608.7 | 590.8 | 500.6 | 449.2 | 1.46× | 1.36× | 1.31× | 1.11× |
| LLaMA2-MoE | 2439.1 | 2152.2 | 1849.2 | 1707.4 | 1468.3 | 1.66× | 1.47× | 1.26× | 1.15× |
| DeepSeek-V2-S | 7233.7 | 5495.1 | 5323.0 | 4958.3 | 4142.4 | 1.74× | 1.33× | 1.28× | 1.20× |

We conduct a detailed analysis on how FlowMoE adapts to heterogeneous GPU clusters. (1) **For clusters with mixed GPU types**, since all-reduce and A2A tasks can only begin once the slowest GPU completes its corresponding computing task (other GPUs will be idle in waiting time after completing their computing tasks), the task timeline for each iteration of FlowMoE and other baselines is determined by the slowest GPU. In this case, FlowMoE can maximize the computing/communication overlap of the slowest GPU, and its performance still outperforms the baselines. To verify this conclusion, we construct a heterogeneous cluster with 16 GPUs on Cluster 1 with different GPU types. We add delay to each iteration on 8 GPUs in one node of Cluster 1 to simulate the computing power reduction. Specifically, we use *register_forward_hook* and *register_hook* in PyTorch to access the forward computing and backward propagation of each layer. Meanwhile, we synchronize the tensor of each layer with *torch.cuda.synchronize* after it completes the forward computing or gradient computing. Then, we obatin the forward computing or backward propagation time of each layer according to the interval between two tensor synchronizations and inject equivalent delays during both passes, which effectively simulates halving the computing power of the GPUs. Table A.12 demonstrates that FlowMoE still obtains the shortest end-to-end training time when the GPUs have different computing power. (2) **For clusters with bandwidth asymmetry**, since AR and A2A tasks are collective communications, all GPUs start and finish communication tasks simultaneously regardless of whether their bandwidths are the same. Therefore, the task scheduling timeline is the same for all GPUs. In this case, FlowMoE still achieves better performance than the baseline due to its efficient pipeline solution.

## K.2 Robustness for Dynamic Hardware Environments

In real large-scale training clusters, the hardware environments including network bandwidth and GPU computing power may change dynamically. To address it, we design the re-Bayesian tuning mechanism for FlowMoE and BO will be automatically re-executed when the hardware environment changes. Specifically, we define a re-execution threshold $\delta$. We denote the iteration time corresponding to the near-optimal $S_p$ predicted by the last BO as $\hat{\mathcal{F}}(S_p^{best})$. If the change of the current iteration time $T$ compared to $\hat{\mathcal{F}}(S_p^{best})$ is more than $\delta$, i.e.,

$$\frac{\left| T - \hat{\mathcal{F}}(S_p^{best}) \right|}{\hat{\mathcal{F}}(S_p^{best})} > \delta, \tag{A.11}$$

the BO will be re-executed to find a new $S_p$.

## K.3 Robustness for Node Dropouts

In FlowMoE, to improve the fault tolerance of the system to handle node dropouts, we adopt the following approach:

First, a replica of each expert parameter is stored separately on two different nodes to ensure that when a node fails, its corresponding expert parameter can still continue to be served by the backup replica. To ensure the parameter consistency between the replicas, we set to synchronize the updating of expert parameters every fixed number of iteration steps (e.g., every 1000 steps) during the training process.

Second, FlowMoE periodically calls *torch.distributed.barrier()* during the training process with a reasonable timeout (e.g., 5 minutes). When a synchronization operation times out and throws an exception, we determine that a node has failed, and obtain the rank of the failed node from the exception information. Then, FlowMoE will execute the following recovery process:

(1) Based on the node replica information, the gating function routing table is updated on all surviving nodes, and the requests originally assigned to the faulty node are remapped to its corresponding backup node;

(2) The remaining surviving nodes are adapted to the new set of nodes and subsequently trained by destroying the current communication group (*dist.destroy_process_group()*) and reinitializing a new distributed communication group (*dist.init_process_group()*);

Therefore, FlowMoE is able to effectively handle the risks from node dropouts in large-scale distributed training, and improves the stability and reliability of the overall training.

