# OpenReview forum: "FlowMoE: A Scalable Pipeline Scheduling Framework for Distributed Mixture-of-Experts Training"
_NeurIPS.cc/2025/Conference — NeurIPS 2025 poster_

### Official Review · Reviewer_GGaM · 2025-06-29

**Clarity:** 4
**Significance:** 3
**Originality:** 3
**Rating:** 5
**Confidence:** 4

**Summary:**

This paper presents a pipeline scheduling framework named FlowMoE for Mixture-of-Expert (MoE) model training. Instead of only pipelinining within the MoE layer, it proposes that 1) other layers including MHA/gating/allreduce/expert layers can also get pipelined in order to achieve higher utilization, and 2) allreduce(AR) ops can be chunked by tensors so as to filling in the gaps between A2A. This paper also presents a BO-based method to find best allreduce chunks and a priority scheduling algorithm for A2A and AR.The evaluations were conducted on a variety of model architectures and the results show convincing speedup over exisiting works.

**Questions:**

1. Could you comment on the GPU utilization when using 1) a smaller batch size, and 2) large number of experts with small number of activated experts?
2. (Section 4.2) Could you clarify if bayesian optimization for AR/A2A scheduling is on the critical path? Can it be made async with respect to training loop in order to further hide the overhead?

**Ethical Concerns:**

["NO or VERY MINOR ethics concerns only"]

**Final Justification:**

The authors have address my concerns and I would like to maintain my score.

**Limitations:**

yes

**Paper Formatting Concerns:**

No.

**Quality:**

3

**Strengths And Weaknesses:**

Strength:
* This paper addresses some less explored areas in MoE training and proposes effective solutions: overlapping for non-MoE layers and priority scheduling of collective communications. The unified pipeline scheduling framework could inspire future work on auto-parallelism for MoE models.
* The authors augment their claims with strong evidence empirically and theoretically, showing effectiveness of their approach.
* This paper is well written and easy to follow.

Weakness:
* It is unclear if microbatching on MHA in forward pass and sharded ARs in backward pass will reduce device utilization. It is my understanding that they will reduce the arithmatic intensity, but the GPU SM utilization (or occupancy) is missing from the evaluation.

---

> ### Author Rebuttal · Authors · 2025-07-27
>
> We thank Reviewer GGaM for the thoughtful feedback and constructive comments. We have responded and addressed every weakness and question in detail. Meanwhile, for weaknesses and questions that have been addressed with additional explanations or experiments in the rebuttal, we will incorporate the corresponding discussions or results into the revised version of the paper. Please see the one-by-one responses below.
>
> >### **[Weakness 1]**
>
> **[Answer 1]** We conducted experiments on Cluster 1 with 16 GPUs to measure GPU utilization under different microbatch sizes. Specifically, we adjusted the microbatch size by changing the pipelining degree $R$ (a larger $R$ results in a smaller microbatch). VanillaEP represents the original mini-batch without pipelining. We used the NVIDIA SMI tool to sample the utilization of all GPUs every 5ms and averaged the values over 1000 iterations. Additionally, we reported the standard deviation of all GPU utilization values to illustrate the degree of data variability. The experimental results are as follows:
>
> |  | Model | $R$ | Average GPU Utilization | Standard Deviation |
> |--------|--------|--------|--------|--------|
> | FlowMoE | GPT2-Tiny-MoE | 2 | 94.96% | 13.35% |
> | FlowMoE | GPT2-Tiny-MoE | 4 | 69.34% | 21.04% |
> | vanillaEP | GPT2-Tiny-MoE | /  | 91.16% | 40.40% |
> | FlowMoE | BERT-Large-MoE | 2 | 96.94% | 3.28% |
> | FlowMoE | BERT-Large-MoE | 4 | 96.14% | 5.91% |
> | vanillaEP | BERT-Large-MoE | /  | 98.67% | 12.59% |
> | FlowMoE | LLaMA2-MoE | 2 | 99.49% | 2.98% |
> | FlowMoE | LLaMA2-MoE | 4 | 98.98% | 2.67% |
> | vanillaEP | LLaMA2-MoE | /  | 99.24% | 15.60% |
> | FlowMoE | DeepSeek-V2-S | 2 | 98.26% | 12.08% |
> | FlowMoE | DeepSeek-V2-S | 4 | 94.93% | 11.10% |
> | vanillaEP | DeepSeek-V2-S | /  | 98.46% | 23.29% |
>
> Based on the above results, we have the following conclusions:
>
> (1) According to the comparison of average GPU utilization, smaller microbatch sizes may result in lower GPU utilization (e.g., when training GPT2-Tiny-MoE with $R=4$ using FlowMoE), but in most cases, GPU utilization is nearly identical to that without pipelining (e.g., when training BERT-Large-MoE, LLaMA2-MoE and DeepSeek-V2-S using FlowMoE). This is because the actual tensors involved in GPU computation remain sufficiently large for larger MoE models, and GPU SM resources are still utilized efficiently. In other words, the microbatches introduced by FlowMoE's pipelining do not waste GPU SM resources for large MoE models;
>
> (2) According to the standard deviation of all GPU utilization values, compared to vanillaEP, FlowMoE maintains relatively stable GPU utilization during training (e.g., the standard deviation is even less than 6% when training BERT-Large-MoE and LLaMA2-MoE). This is because FlowMoE overlaps computing and communication tasks, and the GPU rarely enters idle states waiting for communication. In contrast, vanillaEP's communication tasks must be executed separately, resulting in low GPU utilization during communication task execution.
>
> We will add the above discussion and results on the GPU utilization in the Appendix if the paper is accepted.
>
> >### **[Question 1]**
>
> **[Answer 2]** ●Firstly, smaller batch sizes are more likely to result in lower GPU utilization. We measured GPU utilization with different batch sizes when training different MoE models using FlowMoE on Cluster 1 with 16 GPUs. Similarly, we used the NVIDIA SMI tool to sample the utilization of all GPUs every 5ms and averaged the values over 1000 iterations. The measurement results are as follows:
>
> | Model | Batch Size | Average GPU Utilization |
> |--------|--------|--------|
> | GPT2-Tiny-MoE | 4 | 94.96% |
> | GPT2-Tiny-MoE | 2 | 65.68% |
> | BERT-Large-MoE | 4 | 96.94% |
> | BERT-Large-MoE | 2 | 84.23% |
> | LLaMA2-MoE | 4 | 99.49% |
> | LLaMA2-MoE | 2 | 98.52% |
> | DeepSeek-V2-S | 4 | 98.26% |
> | DeepSeek-V2-S | 2 | 96.28% |
>
> The results demonstrate that smaller batch sizes are more likely to lead to lower GPU utilization, e.g., when training GPT2-Tiny-MoE or BERT-Large-MoE with a batch size of 2. Moreover, GPU utilization remained nearly unchanged when training LLaMA2-MoE or DeepSeek-V2-S. This is because these two MoE models have a larger number of parameters, which keeps the tensor dimensions involved in GPU computation sufficiently large, ensuring that SM resources are still efficiently utilized even with a lower batch size.
>
> ●Secondly, when using a large number of experts with a small number of activated experts, the utilization of GPUs will vary significantly. GPUs hosting frequently activated experts have higher utilization, while those with rarely activated experts experience underutilization.
>
> We also evaluated the impact of the number of activated experts on GPU utilization when using FlowMoE on Cluster 1 with 16 GPUs for a large number of experts. Specifically, we constructed BERT-Large-MoE-w by increasing the number of experts per GPU from 2 to 8 and adjusting the number of activated experts by modifying the capacity factor $f$, where a larger $f$ indicates more uneven token routing by the gate function and fewer activated experts since most tokens are routed to popular experts. The detailed configurations of BERT-Large-MoE-w are as follows:
>
> | MoE Model | # Params (MHA+Gating) | # Params (Experts) | Dataset | L | B | N | M | H | E/P | k |
> |-----------------|----------------------------------|----------------------------|-------------|----|----|----|----|------|---|---|
> | BERT-Large-MoE-w | 25.2M | 3325.9M | wikitext-103 | 24 | 4 | 512 | 512 | 1024 | 8 | 1 |
>
> We used the NVIDIA SMI tool to sample the utilization of each GPU every 5ms and averaged the utilization of each GPU over 1000 iterations. We reported the maximum and minimum GPU utilization in the table below:
>
> | Model | $f$ | Maximum GPU Utilization | Minimum GPU Utilization |
> |--------|--------|--------|--------|
> | BERT-Large-MoE-w | 1.0 | 97.63% | 96.14% |
> | BERT-Large-MoE-w | 4.0 | 98.37% | 56.58% |
> | BERT-Large-MoE-w | 8.0 | 98.92% | 38.16% |
> | BERT-Large-MoE-w | 16.0 | 99.23% | 26.92% |
>
> The above results show that, with a large number of experts, the fewer the number of activated experts, the more unbalanced the computation load on the GPU and the greater the difference in utilization between GPUs.
>
> We will add the above discussion and results on the GPU utilization in the Appendix if the paper is accepted.
>
> >### **[Question 2]**
>
> **[Answer 3]** The Bayesian optimization (BO) process used to adjust the AR chunk size ($S_p$) is not a critical path in the training loop and can be made asynchronous with respect to the training loop. However, BO is crucial to FlowMoE. Meanwhile, FlowMoE's performance is not sensitive to BO hyperparameters, and users do not need to spend time carefully designing these parameters.
>
> Specifically, BO is only executed during the initial profiling phase (typically the first ~80 iterations), during which it continuously suggests and samples a small number of ($S_p$, per-iteration time) pairs (8 by default). Then, BO predicts an approximate optimal $S_p$ based on these sampling pairs and uses it for subsequent training. All training iterations during the initial profiling phase use the existing FlowMoE pipeline and run normally. Therefore, we agree with the reviewer that BO can be redesigned to piggyback on standard iterations (e.g., using a background thread or dedicating evaluation to a shadow process, with BO and the main training loop process exchanging only sampled values and updated $S_p$) without blocking training, thereby further reducing even this minimal overhead. We view this as a valuable direction for future research.

---

> ### Comment · Reviewer_GGaM · 2025-08-04
>
> I'd like to thank the authors for the detailed reply. I also want to point out that `nvidia-smi` tool only reports the percentage of time that *any* SM is active, therefore, it can inflate the result since other SMs can be non-active. To get the accurate GPU utilization that considers all SMs, I'd like to refer the authors to nvidia's DCGM tool and consider revising the evaluation method. Thanks.

---

> ### Author Response · Authors · 2025-08-04
>
> Thank you once again for taking out time to review our paper. We will utilize the limited time remaining in the discussion to make our best effort to remeasure GPU SM utilization using NVIDIA's DCGM tool and attach the results.

---

> ### Author Response · Authors · 2025-08-04
>
> Dear Reviewer GGaM,
>
> Thank you very much for your suggestion. Since the GPU SM utilization measurement function in the NVIDIA DCGM tool is not enabled for consumer-grade GPUs (e.g., RTX series GPUs), we used the CUPTI tool to measure SM utilization. In particular, according to [1-2], DCGM also needs to call CUPTI when measuring SM utilization, therefore CUPTI can obtain more accurate SM utilization. The following three tables show (1) the change in average GPU SM utilization with different microbatch sizes (Table 1), (2) the change in average GPU SM utilization with different batch sizes when using FlowMoE (Table 2), and (3) the maximum and minimum GPU SM utilization for a large number of experts with different numbers of activated experts when using FlowMoE (Table 3) in Cluster 1 with 16 GPUs. Our observations show that although the numerical utilization values measured using different tools may vary, the conclusions obtained remain consistent. We will add this measurement result and the related discussion if accepted.
>
> **Table 1**
> | |Model|$R$|Average GPU SM Utilization|
> |-|-|-|-|
> |FlowMoE|GPT2-Tiny-MoE|2|72.63%|
> |FlowMoE|GPT2-Tiny-MoE|4|48.43%|
> |vanillaEP|GPT2-Tiny-MoE|/|87.09%|
> |FlowMoE|BERT-Large-MoE|2|87.84%|
> |FlowMoE|BERT-Large-MoE|4|78.16%|
> |vanillaEP|BERT-Large-MoE|/|88.90%|
> |FlowMoE|LLaMA2-MoE|2|89.16%|
> |FlowMoE|LLaMA2-MoE|4|88.19%|
> |vanillaEP|LLaMA2-MoE|/|89.49%|
> |FlowMoE|DeepSeek-V2-S|2|89.27%|
> |FlowMoE|DeepSeek-V2-S|4|88.85%|
> |vanillaEP|DeepSeek-V2-S|/|90.77%|
>
> **Table 2**
> |Model|Batch Size|Average GPU SM Utilization|
> |-|-|-|
> |GPT2-Tiny-MoE|4|72.63%|
> |GPT2-Tiny-MoE|2|36.62%|
> |BERT-Large-MoE|4|87.84%|
> |BERT-Large-MoE|2|61.48%|
> |LLaMA2-MoE|4|89.16%|
> |LLaMA2-MoE|2|88.45%|
> |DeepSeek-V2-S|4|89.27%|
> |DeepSeek-V2-S|2|89.06%|
>
> **Table 3**
> |Model|$f$|Maximum GPU SM Utilization|Minimum GPU SM Utilization|
> |-|-|-|-|
> |BERT-Large-MoE-w|1.0|89.20%|87.81%|
> |BERT-Large-MoE-w|4.0|89.72%|50.65%|
> |BERT-Large-MoE-w|8.0|90.30%|31.60%|
> |BERT-Large-MoE-w|16.0|90.68%|19.41%|
>
> **References:**
> [1] NVIDIA Corporation, NVIDIA Developer DCGM Documentation User Guide.
> [2] NVIDIA Corporation, Open Source: DCGM GitHub Repo.

---

### Official Review · Reviewer_YDn2 · 2025-06-30

**Clarity:** 3
**Significance:** 3
**Originality:** 2
**Rating:** 4
**Confidence:** 4

**Summary:**

The paper proposes a scalable framework for scheduling multi-type task pipelines in the context of distributed Mixture-of-Experts training.

**Questions:**

- How does FlowMoE compare with other potential optimization strategies, such as different scheduling algorithms or alternative communication patterns?
- Have the authors considered or compared against other automatic tuning methods beyond Bayesian optimization?
- How sensitive is the performance to the hyperparameters, particularly the BO search space and acquisition function choice?
- Can the authors provide deeper theoretical analysis addressing the convergence properties of the proposed scheduling approach?

**Ethical Concerns:**

["NO or VERY MINOR ethics concerns only"]

**Final Justification:**

The authors have provided a satisfactory rebuttal. As mentioned in my response below, I now consider the work holds research value more than a simple "engineering" tweak. As such, I have upgraded the ratings accordingly.

- Rating: 3 (Borderline Reject) → Revised Score: 4 (Borderline Accept)
- Quality: 2 → 3 (Fair → Good)

**Limitations:**

yes

**Quality:**

3

**Strengths And Weaknesses:**

Strength:
- The paper addresses a well-motivated and practically important problem in distributed MoE training. The authors provide concrete evidence (Table 1) showing that MHA computing, gating, and all-reduce communication constitute 30-40% of iteration time, which existing frameworks neglect.
- The paper provides mathematical formulation of the optimization problem and includes theorems (Theorems 1 and 2) that support the design decisions, particularly for the priority scheduling mechanism.

Weaknesses:
- The core contribution is primarily an engineering optimization that extends existing pipelining techniques. While the unified approach appears valuable, the fundamental algorithmic innovations seem incremental. The engineering contribution, while substantial, is primarily an optimization of existing techniques rather than a fundamental advancement. The impact is thus limited to the specific domain of MoE training optimization.
- While the paper provides mathematical formulation, the theoretical analysis could be deeper. The convergence properties of the proposed scheduling approach are not analyzed, and the Bayesian optimization component appears basic and potentially fragile across various configurations.
- The paper primarily compares against existing MoE frameworks (ScheMoE, Tutel, FasterMoE) and vanilla expert parallelism, but lacks comparison with other recent advances in distributed training optimization.

---

> ### Author Rebuttal · Authors · 2025-07-28
>
> We thank Reviewer YDn2 for the thoughtful feedback and constructive comments. We have responded and addressed every weakness and question in detail. Meanwhile, for weaknesses and questions that have been addressed with additional explanations or experiments in the rebuttal, we will incorporate the corresponding discussions or results into the revised version of the paper. Please see the one-by-one responses below.
>
> >### **[Weakness 1]**
>
> **[Answer 1]** We appreciate the opportunity to clarify the innovation and impact of our work.
>
> ●Firstly, the MoE architecture has become one of the dominant architectures for modern LLMs, including LLaMA4 (Meta, 2025), DeepSeek-V3 (2024), Grok-1 (xAI, 2024), and DBRX (Databricks, 2024). As MoE models become more prevalent,  optimizing their distributed training is both critical and urgent. We believe FlowMoE will make a meaningful contribution to the development of this field.
>
> ●Secondly, existing pipelining techniques focus on overlapping of All-Reduce (AR) and computing, or All-to-All (A2A) and expert computing, but leave two key issues unresolved:  (1) optimal coexistence of AR and A2A, and (2) a unified pipeline scheduling scheme for all major MoE tasks. FlowMoE explicitly addresses these two issues. Therefore, FlowMoE is not an optimization of existing pipeline technologies but proposes a new pipeline acceleration scheme, representing a fundamental innovation.
>
> ●Thirdly, we introduce a formal task-dependency pipelining model and prove that our scheduling strategy strictly reduces or matches per-iteration time compared to sequential or partially pipelined scheduling (see Theorems 1 and 2). This mathematical foundation distinguishes our method from heuristic-driven pipeline strategies, going beyond standard engineering optimization.
>
> ●Fourly, FlowMoE not only focuses on system-level implementation but also emphasizes algorithm design for the pipelining scheme and AR chunk based priority scheduling mechanism (e.g., Algs. 1 and 2 in Sec. 4.2). These contributions clearly extend the impact beyond the specific domain of MoE training optimization.
>
> ●Finally, our submission is highly consistent with the NeurIPS infrastructure track (e.g., libraries, improved implementations and scalability, distributed solutions). FlowMoE provides an efficient distributed solution that improves the implementation of MoE training systems. Additionally, the AI community has been extensively focusing on research to accelerate distributed training through pipelining techniques [1-7], and we believe that FlowMoE's contributions represent a fundamental advancement in this important and evolving field.
>
> >### **[Weakness 2, Question 3, Question 4]**
>
> **[Answer 2]** To deepen the theoretical analysis, we analyze (1) the convergence of our scheduling method and (2) FlowMoE's sensitivity to Bayesian optimization (BO) configurations. Since the above two theoretical analyses also address Questions 3 and 4, we provide a unified response here.
>
> (1) **Convergence of the proposed scheduling method.**
> FlowMoE only changes the execution order of computing and communication tasks without affecting model convergence. We denote the samples processed in each pipeline as one microbatch, and the number of microbatches is equal to the pipelining degree $R$.
>
> Specifically, during the backpropagation of the $l$th transformer block in one iteration, the gradients from all $AT_r^l$ and $E_r^l$ ($1 \le r \le R$) are accumulated and summed. Once the gradient from $AT_1^l$ is also accumulated, the All-Reduce chunk communication task for the MHA and gate is started. Similarly, when the gradient of $E_1^l$ is accumulated, the expert parameters are updated. This effectively prevents the impact of gradient staleness on convergence from early parameter updates. Additionally, to ensure that the accumulated gradients are equivalent with and without pipelining, the loss of each microbatch is scaled by $\frac{1}{R}$, i.e., $\frac{loss^{(r)}}{R}$ ($1 \le r \le R$), and the detailed derivation follows:
>
> ●The loss and gradient when performing backpropagation using the entire mini-batch are expressed as follows:
> $$
> loss=\frac{1}{B}{\textstyle\sum_{i=1}^{B}\ell (x_i,y_i)}, \nabla L_{full}=\nabla\left (\frac{1}{B}{\textstyle\sum_{i=1}^{B}\ell (x_i,y_i)}\right ),\tag{1}
> $$
> which is currently used by all mainstream MoE training frameworks, e.g., Tutel, DeepSpeed-MoE and Megatron-LM.
> ●When dividing mini-batch into $R$ microbatches, the number of samples in each microbatch is $b=\frac{B}{R}$. The loss for each microbatch is:
> $$
> loss^{(r)}=\frac{1}{b}{\textstyle\sum_{i=1}^{b}\ell (x_{r,i},y_{r,i})}.\tag{2}
> $$
> Then, the scaled loss for each microbatch is:
> $$
> \tilde{loss}^{(r)}=\frac{1}{R}\cdot loss^{(r)}=\frac{1}{R}\cdot \frac{1}{b}{\textstyle\sum_{i=1}^{b}\ell (x_{r,i},y_{r,i})}=\frac{1}{B} {\textstyle\sum_{i=1}^{b}\ell (x_{r,i},y_{r,i})}.\tag{3}
> $$
> The cumulative loss of all microbatches is:
> $$
> {\textstyle\sum_{r=1}^{R}\tilde{loss}^{(r)}}={\textstyle\sum_{r=1}^{R}\frac{1}{B}{\textstyle\sum_{i=1}^{b}\ell (x_{r,i},y_{r,i})}}=\frac{1}{B}\textstyle\sum_{r=1}^{R}\textstyle\sum_{i=1}^{b}\ell (x_{r,i},y_{r,i})=\frac{1}{B}\textstyle \sum_{i=1}^{B}\ell (x_{i},y_{i}).\tag{4}
> $$
> According to Eqs. 1 and 4, we can obtain:
> $$
> \nabla\left ({\textstyle\sum_{r=1}^{R}}\tilde{loss}^{(r)} \right )=\nabla\left (\frac{1}{B}{\textstyle\sum_{i=1}^{B}}\ell (x_i,y_i)\right )=\nabla L_{full}.\tag{5}
> $$
> This analysis shows that the gradient update strategy is numerically equivalent with and without pipelining, and FlowMoE does not introduce gradient staleness. The only difference between them is the optimization of task scheduling order during execution, which does not compromise model convergence during training.
>
> (2) **Sensitivity analysis of FlowMoE to BO configurations.**
> FlowMoE is insensitive to BO hyperparameters, which can be manually adjusted.
>
> Firstly, the BO search space should be fixed as (0MB, max tensor size to be communicated per transformer block] to ensure that the optimal AR chunk size is included. Thus, the BO configuration only involves two hyperparameters: the acquisition function and surrogate model. Then, an oversized AR chunk size will affect the prioritization of A2A tasks and may prevent them from being started in time, while an undersized AR chunk size will cause excessive startup overhead. Therefore, there must exist a unique optimal solution that maximizes training speed. For this single-peaked, smooth objective function with a fixed search space, BO consistently converges near the optimum, largely unaffected by hyperparameter variations. The table below shows that different BO settings have minor effect on FlowMoE performance when training BERT-Large-MoE on Cluster 1.
>
> |Acquisition Function|Surrogate Model|Average Per-iteration Time(ms)|
> |-|-|-|
> |EI (0.1)|GPR + Matern|351.9|
> |EI (0.05)|GPR + Matern|358.9|
> |EI (0.2)|GPR + Matern|354.2|
> |Probability of Improvement|GPR + Matern|355.1|
> |Lower Confidence Bound|GPR + Matern|355.4|
> |EI (0.1)|GPR + RBF|357.2|
> |EI (0.1)|GPR + Rational Quadratic|360.2|
>
> \*EI: Expected Improvement
> \*Surrogate Model use Gaussian Process Regression (GPR) with different kernel functions.
>
> We will add the above theoretical analysis in the Appendix if accepted.
>
> >### **[Weakness 3, Question 1]**
>
> **[Answer 3]** ●Firstly, we added a new recent baseline, FSMoE[8]. FSMoE focuses on pipelining for expert computing and communication tasks within the MoE layer, and it maximizes computation-communication overlap within the MoE layer by reasonably scheduling inter-node  and intra-node communication tasks. The table below shows the comparison of average per-iteration time of FlowMoE and FSMoE on Cluster 1.
>
> |Model|# of GPUs|FSMoE|FlowMoE|$S_1$|
> |-|-|-|-|-|
> |GPT2-Tiny-MoE|4|82.8ms|66.1ms|1.25$\times$|
> ||8|98.8ms|87.6ms|1.13$\times$|
> ||16|114.8ms|95.6ms|1.20$\times$|
> |BERT-Large-MoE|4|278.1ms|239.5ms|1.16$\times$|
> ||8|345.1ms|283.2ms|1.22$\times$|
> ||16|421.9ms|351.9ms|1.19$\times$|
> |LLaMA2-MoE|4|928.1ms|763.9ms|1.21$\times$|
> ||8|1110.4ms|906.0ms|1.23$\times$|
> ||16|1292.6ms|1124.0ms|1.15$\times$|
> |DeepSeek-V2-S|4|2096.3ms|1740.8ms|1.20$\times$|
> ||8|2985.4ms|2384.9ms|1.25$\times$|
> ||16|3895.62ms|3205.3ms|1.22$\times$|
>
> \* $S_1$ is the speedups of FlowMoE over FSMoE.
>
> The results show that FlowMoE still achieves better performance gain compared to FSMoE. This is because FlowMoE implements pipeline scheduling for all major MoE-related tasks and introduces BO to adaptively adjust the AR chunk size to maximize computation-communication overlap across the entire transformer block. It is worth noting that the scheduling strategies of FSMoE and FlowMoE are orthogonal and can be combined to further accelerate MoE training.
>
> ●Secondly, Table 9 in Appendix D.2 also shows the superiority of BO over other auto-tuning optimization strategies in FlowMoE.
>
> >### **[Question 2]**
>
> **[Answer 4]** Yes, we have compared BO, Grid Search, and Random Number Generation in Appendix D.2. The results from Table 9 show that using BO to adjust the AR chunk size achieves the shortest per-iteration time among the three methods.
>
> **References:**
> [1] Pipeline Parallelism with Controllable Memory. *NeurIPS*, 2024.
> [2] Rethinking Memory and Communication Costs for Efficient Data Parallel Training of Large Language Models. *NeurIPS*, 2024.
> [3] SAPipe: Staleness-Aware Pipeline for Data Parallel DNN Training. *NeurIPS*, 2022.
> [4] Theoretical Limits of Pipeline Parallel Optimization and Application to Distributed Deep Learning. *NeurIPS*, 2019.
> [5] Pipe-SGD: A Decentralized Pipelined SGD Framework for Distributed Deep Net Training. *NeurIPS*, 2018.
> [6] PipeOffload: Improving Scalability of Pipeline Parallelism with Memory Optimization. *ICML*, 2025.
> [7] Pfeife: Automatic Pipeline Parallelism for PyTorch. *ICML*, 2025.
> [8] FSMoE: A Flexible and Scalable Training System for Sparse Mixture-of-Experts Models. *ASPLOS*, 2025.

---

> > ### Comment · Reviewer_YDn2 · 2025-08-05
> >
> > After carefully considering the authors’ rebuttal, I now appreciate that the main innovation of this paper goes beyond a simple “engineering” tweak. The co-design and integration of scheduling policies for computation and communication across heterogeneous tasks is a meaningful contribution that enables new levels of performance. In the context of the NeurIPS Datasets and Benchmarks track, and for the broader SysML community, I recognize that this type of work holds value.
> >
> > That said, I would encourage the authors to moderate their claims of “fundamental innovation”, as I feel this somewhat overstates the novelty. Similarly, the assertion regarding model convergence could be more precise, rather than “does not affect model convergence”. I would also suggest specifying the empirical evidence supporting this claim.
> >
> > On the experimental side for SOTA comparisons, the authors have strengthened their case by including a new baseline (FSMoE, a concurrent approach from ASPLOS 2025). The results indicate that FlowMoE achieves consistent and notable speedups (1.13×–1.25×) compared to FSMoE across a range of models and GPU configurations. The robustness and necessity of the BO-based auto-tuner are also well-demonstrated.
> >
> > One area that could further improve the paper is the Related Work section. I suggest adding a dedicated subsection to more thoroughly position FlowMoE within the landscape of distributed training research. In particular, it would be helpful to (a) explicitly compare to general-purpose pipeline schedulers such as PipeDream, clarifying key distinctions in terms of scheduling, and (b) discuss orthogonal MoE optimization techniques, for example, NetMoE’s data-placement approach.
> >
> > Overall, the paper has improved and now meets the threshold for Borderline Accept to my opinion, though further clarifications and a more comprehensive discussion of related work would strengthen it even more. I will slightly adjust my ratings accordingly.

---

> > > ### Author Response · Authors · 2025-08-05
> > >
> > > Dear Reviewer YDn2,
> > >
> > > Firstly, thank you very much for recognizing our contributions. We agree with your suggestion and will not directly claim “fundamental innovation,” but will instead focus on emphasizing the novelty of our work and the detailed differences from the baseline.
> > >
> > > Secondly, in the revision of the paper, we will not simply state that “does not affect model convergence”. We will provide theoretical analysis and also present some experimental results to further demonstrate convergence. Regarding experimental evidence, Figure 7 already shows that the final convergence loss of FlowMoE is nearly identical to that of other baselines when training BERT-Large-MoE, and we will also collect convergence evidence for other models. However, due to time constraints, it is difficult to complete convergence experiments within a few days. If the paper is accepted, we will add new convergence experiment results in the final version. We sincerely appreciate your suggestions, as providing both theoretical and experimental results will make FlowMoE's convergence more convincing and persuasive.
> > >
> > > Thirdly, thank you for recognizing our experiments on new SOTA comparisons and BO performance. We will add these results and related discussions to the paper if it is accepted.
> > >
> > > Fourthly, thank you for your suggestions regarding related work. We will add a new subsection to elaborate on FlowMoE's position from two aspects: (1) General pipeline schedulers such as PipeDream and Gpipe enable pipelining by splitting the model across multiple GPUs, and communication between GPUs is required to exchange activation values or gradients across split layers. In contrast, FlowMoE implements pipelining of computing-communication tasks, accelerating model training by hiding communication time as much as possible; (2) discussing orthogonal MoE optimization techniques that can be combined with FlowMoE, including works such as NetMoE, which alleviates communication bottleneck and load imbalance by reducing cross-node token routing and dynamically adjusting expert placement, and works such as FSMoE, which focuses on shortening training time by overlapping inter-node and intra-node communication. We will cite the references and elaborate on the above two aspects if accepted.
> > >
> > > Finally, we sincerely thank you for adjusting the rating and for all your constructive suggestions to strengthen the paper.

---

### Official Review · Reviewer_gKp7 · 2025-07-02

**Clarity:** 3
**Significance:** 3
**Originality:** 3
**Rating:** 5
**Confidence:** 3

**Summary:**

This paper proposed FlowMoE for distributed Mixture-of-Experts (MoE) training that unifies pipeline scheduling across multi-head attention (MHA), gating, expert computation, and both all-to-all (A2A) and all-reduce (AR) communications. By partitioning tensors into micro-batches and introducing a priority-based scheduling mechanism for AR tensor chunks, tuned via Bayesian optimization, FlowMoE maximizes overlap between compute and communication.

**Questions:**

1. Figure 7 shows loss over time but no convergence stability analysis. We understand from other papers that pipelining can sometimes introduce gradient staleness. This is unaddressed in this paper. Is gradient staleness avoided? If there is gradient staleness that can happen, there is a lack of discussion on whether the model performance is affected or not.

2. Energy savings are measured per worker but lack total cluster-wide analysis.  Are their communication costs not covered by the per worker analysis?

**Ethical Concerns:**

["NO or VERY MINOR ethics concerns only"]

**Final Justification:**

Thank you for addressing all the issues we raised during the review process.  We believe the paper is a well written and a would be a good contribution to NeurIPS. We will thus keep our rating of (accept) as is.

**Limitations:**

None.

**Quality:**

3

**Strengths And Weaknesses:**

Strengths:
1. The paper addresses a critical bottleneck in distributed MoE training by pipelining all major tasks (MHA, gating, experts, A2A, all-reduce), unlike prior work that only optimizes MoE-layer tasks. .
2. The design of BO-autotuned tensor partitioning is novel.
3. The proposed design achieve a significant speedup.
4. The approach is backed up with meaningful theory
5. The approach addresses real world issues such as dynamic hardware environments and node dropouts

Weaknesses
1. Experiments use small clusters (max 16 GPUs) and tiny models (e.g., GPT2-Tiny-MoE with 50M expert params). Claims of "trillion-parameter scalability" are unsupported. Also largest model (DeepSeek-V2-S) has only 4 transformer blocks, failing to stress-test the framework. Validate scalability on more devices and larger models will be an added strength.

---

> ### Author Rebuttal · Authors · 2025-07-26
>
> We thank Reviewer gKp7 for the thoughtful feedback and constructive comments. We have responded and addressed every weakness and question in detail except for providing evaluation results for larger-scale GPU clusters. We sincerely apologize for this, because it is very difficult and unrealistic for us to obtain larger-scale hardware resources and measure results within one week. But we will continue to seek larger-scale hardware resource in the future to demonstrate the scalability of FlowMoE. In addition, for weaknesses and questions that have been addressed with additional explanations or experiments in the rebuttal, we will incorporate the corresponding discussions or results into the revised version of the paper. Please see the one-by-one responses below.
>
> >### **[Weakness 1]**
>
> **[Answer 1]** ●Firstly, to stress-test FlowMoE's performance, we measured the training performance of different training frameworks on two scaled-up MoE models (LLaMA2-MoE-L and DeepSeek-V2-M), both of which are close to the memory upper limit of Cluster 1. Specifically, we constructed LLaMA2-MoE-L by increasing the number of transformer blocks in LLaMA2-MoE from 32 to 64. We constructed DeepSeek-V2-M by reducing k to 1 while keeping 7 transformer blocks. The detailed configurations of the two scaled-up MoE models are as follows:
> | MoE Model | # Params (MHA+Gating) | # Params (Experts) | Dataset | L | B | N | M | H | E/P | k | f |
> |-----------------|----------------------------------|----------------------------|-------------|----|----|----|----|------|---|---|---|
> | LLaMA2-MoE-L | 268.4M | 8595.2M | wikitext-103 | 64 | 4 | 512 | 1024 | 4096 | 1 | 1 | 1.0 |
> | DeepSeek-V2-M | 734.3M | 3524.7M | OpenWebText | 7 | 4 | 256 | 5120 | 1536 | 2 | 1 | 1.0 |
>
> Then, the average per-iteration time using different frameworks on these two MoE models is shown as follows (FasterMoE is OOM in any cases):
> | # of GPUs | Model | VanillaEP | Tutel | ScheMoE | FlowMoE | $S_3$ | $S_2$ | $S_1$ |
> |----------------|---------|---------------|--------|---------------|--------------|----------|----------|-----------|
> | 4 | LLaMA2-MoE-L | 2405.1ms | 1927.0ms | 1806.1ms | 1493.8ms | 1.61$\times$ | 1.29$\times$ | 1.21$\times$ |
> | 4 | DeepSeek-V2-M | 535.3ms | 468.4ms | 432.2ms | 352.2ms | 1.52$\times$ | 1.33$\times$ | 1.23$\times$ |
> | 8 | LLaMA2-MoE-L | 2989.1ms | 2493.9ms | 2297.9ms | 1833.8ms | 1.63$\times$ | 1.36$\times$ | 1.25$\times$ |
> | 8 | DeepSeek-V2-M | 944.6ms | 773.4ms | 723.6ms | 552.4ms | 1.71$\times$ | 1.40$\times$ | 1.31$\times$ |
> | 16 | LLaMA2-MoE-L | OOM | OOM | OOM | OOM | / | / | / |
> | 16 | DeepSeek-V2-M | 1254.6ms | 956.9ms | 893.4ms | 708.8ms | 1.77$\times$ | 1.35$\times$ | 1.26$\times$ |
>
> \* OOM: out-of-memory
> \* $S_1$, $S_2$ and $S_3$ are the speedups of FlowMoE over ScheMoE, Tutel and vanillaEP, respectively.
>
> The results above show that FlowMoE still achieved the best training performance among all baselines on larger models.
>
> ●Secondly, verifying the scalability of FlowMoE on a larger-scale GPU cluster would indeed be an added strength. However, we sincerely apologize that obtaining a larger-scale hardware resource and evaluating the results within a week is very difficult and unrealistic for us. Nevertheless, our current clusters and models are already sufficient to demonstrate the effectiveness of FlowMoE's pipeline mechanism. We will continue to seek larger-scale hardware resource in the future to demonstrate the scalability of FlowMoE.
>
> ●Thirdly, we agree with the reviewer that any claim of “trillion-parameter scalability” is indeed inappropriate in our paper since we lack experiments on trillion-parameter models. The sentence: "The parameter size of modern large language models (LLMs) can be scaled up to the trillion-level via the sparsely-activated Mixture-of-Experts (MoE) technique to avoid excessive increase of the computational costs", we had in the abstract may have caused misunderstanding. We apologize for that. Thereby, we will modify this sentence to: "The parameter size of modern large language models (LLMs) can be scaled up to a massive scale using the sparsely-activated Mixture-of-Experts (MoE) technique to avoid excessive increases in computational costs." to rectify this issue.
>
> >### **[Question 1]**
>
> **[Answer 2]** Yes, FlowMoE avoids the stale gradient updating problem, and FlowMoE's scheduling mechanism only changes the execution order of computing and communication tasks without affecting the convergence of the model.
>
> We denote the samples processed in each pipeline as one microbatch, and the number of microbatches is equal to the pipelining degree $R$.
>
> Specifically, during the backpropagation of the $l$th transformer block in one iteration, the gradients from all $AT_r^l$ and $E_r^l$ ($1 \le r \le R$) after backpropagation are accumulated and summed. Once the gradient from $AT_1^l$ is also accumulated, the All-Reduce chunk communication task for the MHA and gate function in the $l$th transformer block is started. Similarly, when the gradient of $E_1^l$ is accumulated, the expert parameters are updated. This effectively prevents the parameters of MHA, the gate function, and the expert from being updated early, thereby avoiding gradient staleness. Additionally, to ensure that the accumulated gradients are equivalent with and without pipelining, we scale the loss calculated for each microbatch by $R$, i.e., $\frac{loss^{(r)}}{R}$ ($1 \le r \le R$), where $loss^{(r)}$ is the loss calculated using the samples from the $r$th microbatch, and the detailed theoretical derivation is as follows:
>
> ●The loss and gradient when performing backpropagation using the entire mini-batch are expressed as follows:
> $$
> loss=\frac{1}{B}{\textstyle \sum_{i=1}^{B} \ell (x_i,y_i)}, \nabla L_{full}=\nabla \left (\frac{1}{B}{\textstyle \sum_{i=1}^{B}\ell (x_i,y_i)}\right ),\tag{1}
> $$
> which is currently used by all mainstream MoE training frameworks [1,2,3].
> ●When dividing mini-batch into $R$ microbatches, the number of samples in each microbatch is $b=\frac{B}{R}$. The loss for each microbatch is:
> $$
> loss^{(r)}=\frac{1}{b}{\textstyle \sum_{i=1}^{b}\ell (x_{r,i},y_{r,i}) }.\tag{2}
> $$
> Then, the scaled loss for each microbatch is:
> $$
> \tilde{loss}^{(r)}=\frac{1}{R}\cdot loss^{(r)}=\frac{1}{R}\cdot \frac{1}{b}{\textstyle \sum_{i=1}^{b}\ell (x_{r,i},y_{r,i}) }=\frac{1}{B} {\textstyle \sum_{i=1}^{b}\ell (x_{r,i},y_{r,i})}.\tag{3}
> $$
> The cumulative loss of all microbatches is:
> $$
>  {\textstyle \sum_{r=1}^{R}\tilde{loss}^{(r)}} =  {\textstyle\sum_{r=1}^{R}\frac{1}{B} {\textstyle \sum_{i=1}^{b}\ell (x_{r,i},y_{r,i})}}=\frac{1}{B}\textstyle\sum_{r=1}^{R}\textstyle \sum_{i=1}^{b}\ell (x_{r,i},y_{r,i})= \frac{1}{B}\textstyle \sum_{i=1}^{B}\ell (x_{i},y_{i}).\tag{4}
> $$
> According to Eqs. 1 and 4, we can obtain:
> $$
> \nabla \left (  {\textstyle \sum_{r=1}^{R}}\tilde{loss}^{(r)} \right )= \nabla \left (\frac{1}{B} {\textstyle \sum_{i=1}^{B}}\ell (x_i,y_i)\right )=\nabla L_{full}.\tag{5}
> $$
> This analysis shows that the gradient update strategy is numerically equivalent with and without pipelining, thereby FlowMoE does not introduce gradient staleness. The only difference between them is the optimization of task scheduling order during execution, which does not compromise model convergence during training. We will add this discussion in the Appendix if accepted.
>
> >### **[Question 2]**
>
> **[Answer 3]** Please let us clarify that our measurement process actually does cover the entire cluster-wide energy usage, including both the communication cost and computing cost.
>
> We use the NVIDIA SMI tool to sample the real-time power of each GPU in the cluster every 5ms, then integrate these sampled data over time to calculate the energy consumption of each GPU during the training process. We sum the energy consumption of all GPUs to obtain the total energy consumption, then divide it by the number of GPUs and the number of iterations to obtain the average per-worker (per-GPU) energy consumption as shown in Table 6. It is worth noting that the power reported by the NVIDIA SMI tool includes both communication and computing costs. According to the official documentation of NVIDIA SMI [4], the energy consumption measured by the NVIDIA SMI tool covers the entire GPU card's power consumption, where communication power includes (1) PCIe communication costs, (2) NVLink/NVSwitch communication costs, and (3) collective communication costs (e.g., NCCL All-Reduce and All-to-All). In addition, some representative works [5,6] also used the NVIDIA SMI tool to monitor GPU communication cost. Therefore, our measurement results effectively reflect the total cluster-wide energy usage, compatibly providing a fair comparison with the literature.
>
> **References:**
> [1] Tutel: Adaptive Mixture-of-Experts at Scale. *MLSys*, 2023.
> [2] DeepSpeed-MoE: Advancing Mixture-of-Experts Inference and Training to Power Next-Generation AI Scale. *ICML*, 2022.
> [3] Megatron-LM: Training Multi-Billion Parameter Language Models Using Model Parallelism. *arXiv:1909.08053*, 2019.
> [4] NVIDIA Corporation, *NVIDIA System Management Interface (nvidia-smi) User Guide*.
> [5] Evaluating Modern GPU Interconnect: PCIe, NVLink, NV-SLI, NVSwitch and GPUDirect. *IEEE TPDS*, 2019.
> [6] AUGEM: Automatically Generate High Performance Dense Linear Algebra Kernels on x86 CPUs. *SC*, 2013.

---

> > ### Comment · Reviewer_gKp7 · 2025-08-07
> > **Response to Rebuttal**
> >
> > Thank you for addressing the issues raised in our review in your rebuttal.
> >
> > Weakness 1: I appreciate the added experiments. Also, I suggest (1) clarify in the paper that current results hold up to 8 GPUs and explicitly state the OOM limit at 16 GPUs, (2) replace “massive scale” with the concrete model sizes you’ve demonstrated.
> >
> > Question 1: Thanks for the derivation. I think including a convergence plot comparing FlowMoE to a standard baseline in Appendix will be a strength.
> >
> > Question 2: Your answer is reasonable.
> >
> > We believe our rating of (accept) is well justified and keep it as is.

---

> > > ### Author Response · Authors · 2025-08-08
> > >
> > > Dear Reviewer gKp7,
> > >
> > > Thank you very much for taking out time to review our paper and for maintaining the accept rating.
> > >
> > > We appreciate your constructive suggestions:
> > >
> > > ●We will clarify in the paper that the above partial results for larger-scale models hold up to 8 GPUs and explicitly state the OOM limitation of 16 GPUs.
> > >
> > > ●We will replace "massive scale" with the concrete model sizes we have demonstrated.
> > >
> > > ●We will add a convergence plot comparing FlowMoE to a standard baseline in the revision of the paper if accepted.
> > >
> > > We believe these changes will further strengthen the quality of our paper. Thank you again for your valuable suggestions.

---

### Official Review · Reviewer_hxBa · 2025-07-05

**Clarity:** 3
**Significance:** 3
**Originality:** 3
**Rating:** 4
**Confidence:** 4

**Summary:**

The paper proposes FlowMoE, a scalable pipeline scheduling framework to improve the training efficiency of distributed Mixture-of-Experts (MoE) models. Unlike prior MoE frameworks that focus primarily on pipelining within the MoE layer, FlowMoE constructs a unified pipeline that includes Multi-Head Attention (MHA) computing, gating, expert computing, and All-to-All (A2A) communication, and additionally pipelines all-reduce communication using a tensor chunk-based priority scheduling mechanism. It leverages Bayesian Optimization (BO) to automatically tune the all-reduce partition size for optimal overlap, aiming to reduce iteration time, energy consumption, and memory usage during training. Extensive experiments on two GPU clusters with real-world MoE models (e.g., GPT2-Tiny-MoE, LLaMA2-MoE, BERT-Large-MoE, DeepSeek-V2-S) and 675 customized MoE layers demonstrate 14%-57% speedup, 10%-39% energy reduction, and 7%-32% memory savings over existing state-of-the-art MoE frameworks such as ScheMoE, Tutel, and FasterMoE.

**Questions:**

* Have you observed any cases where FlowMoE's aggressive overlap leads to slower per-iteration time on specific settings?
* Could you elaborate on how FlowMoE adapts to heterogeneous GPU clusters (e.g., mixed GPU types or bandwidth asymmetry) and whether BO retuning is required for every hardware change? How sensitive is FlowMoE’s performance to the BO process?

**Ethical Concerns:**

["NO or VERY MINOR ethics concerns only"]

**Final Justification:**

The authors have address my concerns and I would like to maintain my score.

**Limitations:**

Yes

**Paper Formatting Concerns:**

I don't have any paper formatting concerns.

**Quality:**

3

**Strengths And Weaknesses:**

# Strengths:

* The proposed FlowMoE addresses a practical bottleneck in training large-scale MoE models, which is highly relevant as LLM scaling continues.
* The paper carefully formalizes pipeline scheduling and priority mechanisms, and integrates BO for adaptive optimization, reducing manual tuning.
* The experiments are comprehensive, with detailed comparisons under diverse settings.

# Weaknesses:

* While the paper claims novelty in pipelining MHA computing, gating, expert computing, A2A, and all-reduce communication together, it is unclear how fundamentally novel this scheduling is compared to prior works [1-3].
* While BO tuning of the all-reduce chunk size improves performance, the paper does not provide insights into why the BO-tuned configurations lead to specific speedups compared to baselines or how sensitive the system is to these hyperparameters beyond empirical gains.

[1] Chen, Chang, et al. "Centauri: Enabling efficient scheduling for communication-computation overlap in large model training via communication partitioning." Proceedings of the 29th ACM International Conference on Architectural Support for Programming Languages and Operating Systems, Volume 3. 2024.

[2] Jangda, Abhinav, et al. "Breaking the computation and communication abstraction barrier in distributed machine learning workloads." Proceedings of the 27th ACM International Conference on Architectural Support for Programming Languages and Operating Systems. 2022.

[3] Wang, Shibo, et al. "Overlap communication with dependent computation via decomposition in large deep learning models." Proceedings of the 28th ACM International Conference on Architectural Support for Programming Languages and Operating Systems, Volume 1. 2022.

---

> ### Author Rebuttal · Authors · 2025-07-25
>
> We thank Reviewer hxBa for the thoughtful feedback and constructive comments. We have responded and addressed every weakness and question in detail. Meanwhile, for weaknesses and questions that have been addressed with additional explanations or experiments in the rebuttal, we will incorporate the corresponding discussions or results into the revised version of the paper. Please see the one-by-one responses below.
>
> >### **[Weakness 1]**
>
> **[Answer 1]** Prior works [1–3] focus on non-MoE models and primarily consider pipelining between computing and All-Reduce (AR) communication. In contrast, FlowMoE is specifically designed for MoE models, where All-to-All (A2A) communication is introduced due to the sparsely activated expert layers. Compared to works [1-3], FlowMoE has two fundamental novelties:
> (1) the co-existence and scheduling of both AR and A2A communication tasks, and
> (2) the pipelining of computing tasks (MHA, gating and expert) with two heterogeneous communication tasks (AR and A2A).
> In particular, to the best of our knowledge, FlowMoE is the first work to pipeline these five tasks together.
>
> In addition, as shown in Fig. 2b, in the backpropagation of the MoE model, the expert computing of the $l$th transformer block depends on the combining A2A of the $l$th transformer block, the dispatch A2A of the $l$th transformer block depends on the expert computing of the $l$th transformer block, and the MHA computing of the $(l+1)$th transformer block depends on the dispatch A2A of the $l$th transformer block. However, the AR communication of the $l$th transformer block only depends on the $l$th transformer block MHA computing and gating. Therefore, the A2A communication is not equivalent to the AR communication. Works [1-3] do not consider these complex dependencies and cannot be directly used to solve the pipeline problem of distributed MoE training. The scheduling mechanism of FlowMoE is essentially an efficient solution for scheduling multi-type tasks in distributed MoE training.
>
> >### **[Weakness 2]**
>
> **[Answer 2]** BO auto-tuning is an indispensable part of FlowMoE and is absolutely necessary. While FlowMoE's efficient performance depends on the introduced BO process (Table 10 in Appendix D.2 demonstrates this conclusion), but it is less sensitive to the hyperparameters used in BO, which are of lower importance and can be manually adjusted.
>
> Firstly, regarding your point on providing insights, we clarify that: An oversized All-Reduce (AR) chunk size will affect the prioritization of All-to-All (A2A) tasks and may prevent them from being started in time. In contrast, an undersized AR chunk size will result in excessive startup overhead. Therefore, neither an oversized nor undersized AR chunk size can achieve the shortest per-iteration time, and there must exist a unique optimal solution that maximizes training speed. The BO optimizer balances these two competing effects by sampling and learning from actual iteration times, efficiently finding an AR chunk size​ that maximizes overlap without incurring excessive overhead. This tuning is crucial because the optimal trade-off point varies across models, GPU interconnect topologies, and MoE configurations. We elaborate on these in Appendix D and Theorem 2, where we show theoretically (and empirically in Table 9/10 and Fig. 4) that BO-tuned configurations consistently reduce iteration time compared to fixed or grid-based alternatives. We will clarify this explanation further in the main text if accepted.
>
> Then, in FlowMoE, the objective function for per-iteration time regarding the AR chunk size is a single-peaked, smooth objective function with a fixed search space. For this objective function, the process of using BO to find the optimal solution is insensitive to the two hyperparameters (acquisition function and surrogate model) of BO, and BO always converges and approaches the optimal solution. The following table shows the comparison results of average per-iteration time when training BERT-Large-MoE with different BO parameter configurations on Cluster 1. The results indicate that BO hyperparameters have a minor impact on FlowMoE performance, and different BO configurations lead to similar iteration time.
>
> |Acquisition Function|Surrogate Model|Average Per-iteration Time(ms)|
> |-|-|-|
> |EI (0.1)|GPR + Matern|351.9|
> |EI (0.05)|GPR + Matern|358.9|
> |EI (0.2)|GPR + Matern|354.2|
> |Probability of Improvement|GPR + Matern|355.1|
> |Lower Confidence Bound|GPR + Matern|355.4|
> |EI (0.1)|GPR + RBF|357.2|
> |EI (0.1)|GPR + Rational Quadratic|360.2|
>
> \*EI: Expected Improvement
> \*Surrogate Model use Gaussian Process Regression (GPR) with different kernel functions.
>
> >### **[Question 1]**
>
> **[Answer 3]** No, FlowMoE will not obtain slower per-iteration time in any case. We analyzed three cases:
>
> (1) **The communication task time is much longer than the computing task time.** When the communication time is very long, the time of the All-to-All (A2A) communication task alone completely covers the computing task time. In this case, the All-Reduce (AR) chunk based priority scheduling mechanism will fail because the AR chunk cannot be inserted into the A2A communication task. The performance of FlowMoE will be the same as ScheMoE, Tutel, and FasterMoE, but better than vanillaEP due to the hidden computing task time.
>
> (2) **The computing task time is much longer than the communication task time.** When the computing time is very long, all the communication task time can be covered by the computing task time. In this case, FlowMoE will outperform ScheMoE, Tutel, and FasterMoE due to the hidden AR task time, and better than vanillaEP because it further hides the A2A task time.
>
> (3) **The communication task time is comparable to the computing task time.** In this case, FlowMoE outperforms all baselines because it maximizes the overlap of multi-type tasks.
>
> In summary, in all cases, the performance of FlowMoE is greater than or equal to that of ScheMoE, Tutel, and FasterMoE, and is always better than that of vanillaEP.
>
> >### **[Question 2]**
>
> **[Answer 4]** ●Firstly, we conducted a detailed analysis on how FlowMoE adapts to heterogeneous GPU clusters. (1) For clusters with mixed GPU types, since All-Reduce (AR) and All-to-All (A2A) tasks can only begin once the slowest GPU completes its corresponding computing task (other GPUs will be idle in waiting time after completing their computing tasks), the task timeline for each iteration of FlowMoE and other baselines is determined by the slowest GPU. In this case, FlowMoE can maximize the computing/communication overlap of the slowest GPU, and its performance still outperforms the baselines. Table 12 in Appendix G.1 also demonstrates the same conclusion when comparing different frameworks on GPU clusters with heterogeneous computing power. (2) For clusters with bandwidth asymmetry, since AR and A2A tasks are collective communications, all GPUs start and finish communication tasks simultaneously regardless of whether their bandwidths are the same. Therefore, the task scheduling timeline is the same for all GPUs. In this case, FlowMoE still achieves better performance than the baseline due to its efficient pipeline solution.
>
> ●Secondly, it is not necessary to conduct the BO retuning every time there is a hardware change. We have designed a re-Bayesian tuning mechanism for FlowMoE in Appendix G.2. Specifically, we define a re-execution threshold $\delta$. We denote the iteration time corresponding to the near-optimal AR chunk size $S_p$ predicted by the last BO as $\hat{\mathcal{F}}(S_p^{best})$. If the change of the current iteration time $T$ compared to $\hat{\mathcal{F}}(S_p^{best})$ is more than $\delta$, i.e.,
> $$
> \frac{\left | T-\hat{\mathcal{F}}(S_p^{best}) \right |}{\hat{\mathcal{F}}(S_p^{best})}> \delta,
> $$
> the BO will be re-executed to find a new $S_p$. This is based on the understanding that there is only one optimal AR chunk size to minimize the iteration time for the same hardware cluster (see **[Answer 2]** for detailed analysis). Therefore, when $T$ and $\hat{\mathcal{F}}(S_p^{best})$ are close, it indicates that the hardware conditions have not changed or have only slightly fluctuated, and there is no need to readjust the AR chunk size. However, when $T$ and $\hat{\mathcal{F}}(S_p^{best})$ differ significantly, it indicates that the hardware conditions have changed significantly, and the optimal AR chunk size has also changed. At this point, the BO needs to be re-executed to update the AR chunk size.
>
> ●Thirdly, FlowMoE's performance is sensitive to the BO process. Table 10 in Appendix D.2 compares the per-iteration time of FlowMoE using BO auto-tuning and different fixed AR chunk sizes. The results show that randomly selected fixed AR chunk sizes are likely to severely limit FlowMoE's performance. Therefore, BO auto-tuning is an indispensable part of FlowMoE and can maximize its performance. Although FlowMoE's performance is sensitive to the BO process, it is insensitive to the hyperparameters used in BO and they can be manually adjusted (see **[Answer 2]** for detailed analysis).

---

> ### Comment · Reviewer_hxBa · 2025-08-03
> **Response**
>
> Thank you for your rebuttal. Some of my concerns (i.e., BO tuning) are addressed, but I still think the novelty of this paper is limited given that there're lots of related works in this filed.
>
> First, it's true that some [1–3] of prior works focus on non-MoE models. But it's unclear to me why the pipelining of computing tasks (MHA, gating and expert) with communication tasks (AR and A2A) is fundamentally different and complex, given this sequential execution dependency. Second, there are also some prior works mainly focusing on computation-communication overlapping for MoE models, for example:
>
> * Zhang, Shulai, et al. Comet: Fine-grained computation-communication overlapping for mixture-of-experts. https://arxiv.org/pdf/2502.19811v1
> * Jiang, Chenyu, et al. Lancet: Accelerating mixture-of-experts training via whole graph computation-communication overlapping. MLSys 2024
> * Punniyamurthy, Kishore, Khaled Hamidouche, and Bradford M. Beckmann. Optimizing distributed ml communication with fused computation-collective operations. SC 2024.
>
> Therefore, I will maintain my score.

---

> > ### Author Response · Authors · 2025-08-03
> >
> > Thank you once again for taking out time to review our paper. We hope that our rebuttal satisfactorily addressed all your concerns.

---

> > ### Author Response · Authors · 2025-08-05
> >
> > Dear Reviewer hxBa,
> >
> > Thank you very much for your response. Please allow us to clarify our novelty. Firstly, the pipelining in non-MoE models only involves the overlap between AR and computing tasks, and only needs to consider AR-computing dependency and computing-computing dependency. The pipelining in MoE models involves two different types of collective communication, and requires additional consideration of (1) A2A-computing dependency and (2) the optimal coexistence of AR and A2A, which makes the pipelining of all major MoE tasks more complex and not a simple extension of traditional pipelining methods.
> >
> > Secondly, FlowMoE is fundamentally different from existing work on computing-communication overlapping in MoE models. Zhang et al. focused only on the MoE layer, i.e., implementing A2A-computing overlapping, without considering AR-computing overlapping and AR-A2A coexistence. Jiang et al. determined the scheduling order of different tasks by constructing a computing-communication DAG, without considering microbatch-level partitioning of the same task. Punniyamurthy et al. focuses on the fusion execution of computing-communication operations and reducing kernel-level startup overhead. However, it does not consider the scheduling relationships among all major MoE tasks or how to maximize computing-communication overlapping. It is worth noting that FlowMoE is orthogonal to the work of Jiang et al. and Punniyamurthy et al. and can be applied on top of them.
> >
> > We will enrich our literature review adding a more detailed discussion of the differences between FlowMoE and existing pipeline methods including the above references, in the revision of the paper if accepted.

---

### Note · Authors · 2025-08-11

Dear chairs and reviewers,

Thank you very much for all your efforts in the review and discussion. We sincerely hope that our work can be accepted and contribute to the field of distributed MoE training. We have summarized all reviewers’ comments and our responses as follows:

● To address comments/concerns by Reviewer hxBa: Firstly, we addressed concerns about BO tuning and clarified FlowMoE’s insensitivity to BO configurations, to the reviewer’s satisfaction. Secondly, regarding the novelty, our original expression may have been unclear. We attempted to re-write this part. Hopefully, this better highlights and clarifies our novelty/contributions to the reviewer's satisfaction. Specifically, FlowMoE addresses the unified scheduling across all major tasks and the optimal coexistence of heterogeneous communication tasks in MoE training, proposing a mathematically proven novel pipeline acceleration framework that represents a substantial advancement of distributed solutions beyond simple extension of traditional pipeline methods. FlowMoE can be combined with many orthogonal optimization works and promote distributed pipeline optimization to a new stage.

● To address comments/concerns by Reviewer gKp7: We addressed concerns about larger-scale model validation, convergence, and energy measurements, to the reviewer’s satisfaction. We clarified that FlowMoE introduces no gradient staleness and that our reported energy measurements covers the entire cluster-wide usage.

● To address comments/concerns by Reviewer YDn2: We addressed the reviewer's concerns about innovation, convergence, BO robustness, and SOTA comparison, to the reviewer's satisfaction. The reviewer has appreciated that the main innovation of this paper goes beyond a simple "engineering" tweak and recognized its contribution and value. We also clarified that FlowMoE's gradient update strategy is numerically equivalent to that without pipelining and that FlowMoE is insensitive to BO configurations.

● To address comments/concerns by Reviewer GGaM: We addressed the reviewer's concerns about GPU utilization, to the reviewer's satisfaction. We provided precise GPU SM utilization using the new tool recommended by the reviewer. The results show that GPU SM resources are effectively utilized in FlowMoE for large MoE models.

Thanks again for your suggestions. If accepted, we will incorporate all our responses and results into the revision to further strengthen the paper.

---

### Decision · Program_Chairs · 2025-09-17

**Decision:**

Accept (poster)

**Comment:**

Overall, the paper will be a good contribution to NeurIPS which is also a consensus among the reviewers.

(a) Summary of Claims

The paper proposes FlowMoE, a pipeline scheduling framework for distributed MoE training. It unifies scheduling of different components such as MHA, gating, expert computation, A2A, and AR communication, and introduces a tensor chunk-based priority mechanism tuned via Bayesian Optimization (BO). The authors show empirical speedup, energy savings, and memory reduction compared to different MoE frameworks.

(b) Strengths

- Tackles a practical bottleneck in MoE training with a unified and adaptive scheduling framework.
- Strong empirical results.
- Theoretical backing (Theorems 1 & 2) and robustness to BO hyperparameters.

(c) Weaknesses

- Novelty is limited; some reviewers view it as an engineering extension of prior work.
- Experiments are limited to 16 GPUs and sub-trillion parameter models, despite scalability claims.
- GPU utilization initially measured with less precise tools (later addressed).

(d) Justification for Recommendation

Despite concerns about novelty, the paper presents an important system contribution. In particular, the unified scheduling of heterogeneous tasks and integration of BO tuning are valuable. Therefore, this is a good contribution to NeurIPS.

(e) Rebuttal Summary and Discussion

Reviewer hxBa maintained a borderline accept, citing limited novelty even with satisfactory responses on BO tuning.
Reviewer gKp7 appreciated the added experiments and theoretical clarifications, maintaining an accept.
Reviewer YDn2 upgraded to borderline accept after recognizing the co-design of scheduling policies as a meaningful contribution.
Reviewer GGaM accepted the paper and appreciated the authors’ follow-up with more accurate GPU utilization metrics.